# Degron-tagged reporters probe membrane topology and enable the specific labelling of membrane-wrapped structures

Katharina B. Beer [1,4], Gholamreza Fazeli[1,4], Kristyna Judasova[2,3], Linda Irmisch[1], Jona Causemann[1], Jörg Mansfeld [3] & Ann M. Wehman [1]

Visualization of specific organelles in tissues over background fluorescence can be challenging, especially when reporters localize to multiple structures. Instead of trying to identify proteins enriched in specific membrane-wrapped structures, we use a selective degradation approach to remove reporters from the cytoplasm or nucleus of *C. elegans* embryos and mammalian cells. We demonstrate specific labelling of organelles using degron-tagged reporters, including extracellular vesicles, as well as individual neighbouring membranes. These degron-tagged reporters facilitate long-term tracking of released cell debris and cell corpses, even during uptake and phagolysosomal degradation. We further show that degron protection assays can probe the topology of the nuclear envelope and plasma membrane during cell division, giving insight into protein and organelle dynamics. As endogenous and heterologous degrons are used in bacteria, yeast, plants, and animals, degron approaches can enable the specific labelling and tracking of proteins, vesicles, organelles, cell fragments, and cells in many model systems.

[1] Rudolf Virchow Center, Julius-Maximilians-Universität Würzburg, Josef-Schneider-Str. 2, 97080 Würzburg, Germany. [2] Dresden International Graduate School for Biomedicine and Bioengineering (DIGS-BB), Dresden, Germany. [3] Cell Cycle, Biotechnology Center, Technische Universität Dresden, Tatzberg 47-49, 01307 Dresden, Germany. [4] These authors contributed equally: Katharina B. Beer, Gholamreza Fazeli. Correspondence and requests for materials should be addressed to A.M.W. (email: ann.wehman@uni-wuerzburg.de)

Membranes form barriers that separate cells from their environment and separate diverse subcellular compartments so that they can carry out distinct functions within cells[1]. Membranes are also highly dynamic, undergoing fusion and fission events during endocytosis, ectocytosis, and cytokinesis[2,3]. As membrane bilayers are less than 10 nm in diameter, light microscopic techniques struggle to distinguish neighbouring membranes or to tell when membranes have fused to have a closed topology[4]. Therefore, techniques that allow specific labelling of distinct structures are required to visualize membrane dynamics and probe membrane topology in living cells.

Fluorescent reporters binding specific phosphatidylinositol species are popular for studying membrane dynamics[5], but these lipids are often present on multiple structures, making it hard to distinguish individual organelles. One example is during phagocytosis, when cellular debris or cell corpses are engulfed by the plasma membrane[6]. Both the corpse and engulfing cell plasma membranes contain the same phosphatidylinositol species, making it challenging to distinguish these membranes in living cells. Electron microscopy and super-resolution light microscopy can visualize the few tens of nm that separate the phagosome membrane from the corpse membrane[7], but these techniques rely on fixation, which makes it challenging to study dynamics. Another example is extracellular vesicles (EVs) released from cells. The content of vesicles that bud from the plasma membrane by ectocytosis is similar to the membrane and associated cytoplasm from which they originate[8], which makes it hard to distinguish released vesicles from the plasma membrane of neighbouring cells. The lack of specific markers for EVs using conventional reporters has limited our understanding of their cell biology[9]. Thus, new approaches are needed to label specific membranes.

In order to develop reporters that visualize specific membrane structures, we repurposed the cell's endogenous machinery for selective degradation. We were inspired by protease protection assays using exogenous proteases[10], but wanted to establish an in vivo system that did not require detergent-mediated permeabilization of the plasma membrane. Degrons are degradation motifs that target specific proteins for ubiquitination and degradation, which has led to degron tagging being used as an alternative loss-of-function approach to RNA interference or genetic knockouts[11]. Degrons recruit ubiquitin ligases to polyubiquitinate target proteins, resulting in the proteasomal degradation of cytosolic targets or the lysosomal degradation of transmembrane targets[12]. Rather than using degron tags for a loss-of-function technique, we used the degradation of degron-tagged reporters in the cytosol to specifically label certain cells, cell fragments, organelles, and vesicles.

To test endogenous degrons, we primarily used the zinc finger 1 (ZF1) degron from the PIE-1 protein to degrade fluorescent reporters in developing C. elegans embryos. The ZF1 degron is a 36 amino acid motif recognized by the SOCS-box protein ZIF-1, which binds to the elongin C subunit of an ECS ubiquitin ligase complex[13]. ZIF-1 is expressed in sequential sets of differentiating somatic cells[14], resulting in a stereotyped pattern of degradation in developing embryos[13] (Fig. 1a). Fusing the ZF1 degron to a target protein results in degradation within 30–45 min of ZIF-1 expression in both embryonic and adult tissues[15]. As an alternate approach, we used the C-terminal phosphodegrons (CTPD) from the C. elegans OMA-1 protein[16]. Two threonines in the C-terminus of OMA-1 are phosphorylated after fertilization, leading to recognition of OMA-1 by multiple SCF ubiquitin ligase complexes and rapid proteasomal degradation in embryos at the end of the one-cell stage[16,17] (Fig. 1b). We also used a heterologous degron in mammalian cells, the auxin-inducible degron (AID) from plants[18]. The 68 amino acid AID motif from IAA17 is

recognized by the F-Box protein TIR1 in the presence of the auxin family of plant hormones[19]. Auxins are cell permeable and TIR1 is able to become part of endogenous SCF ubiquitin ligase complexes in model systems from yeast to mammals in order to ubiquitinate AID-tagged proteins[18]. The AID system is thus a three-component system, allowing temporal control of degradation by addition of auxin hormone and spatial control of degradation by the expression of TIR1 in different cells (Fig. 1c).

Here, we show that degron-tagged reporters separated from the ubiquitin ligase complex by intervening membranes are no longer accessible to ubiquitination and degradation in C. elegans embryos or mammalian cells. This results in background-free labelling of specific cells, organelles, and vesicles. This improvement in the signal-to-noise ratio enables the visualization of EVs in vivo, the long-term tracking of individual phagosomes, as well as distinguishing a corpse plasma membrane from the engulfing phagosome membrane in vivo. In addition, degron tagging allows us to measure the timing of nuclear envelope breakdown (NEBD) and abscission during cell division. Degron-tagged reporters thus provide a convenient method for investigating in vivo dynamics from the level of proteins to cells.

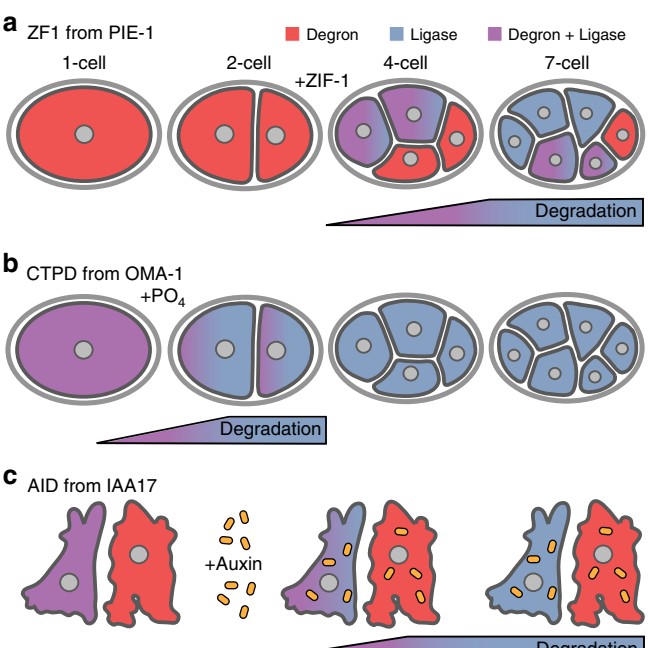

**Fig. 1** Comparison of degron approaches. **a** Proteins with a zinc finger 1 degron (ZF1, red) are stable before expression of the ubiquitin ligase adaptor ZIF-1 (blue) in C. elegans embryos. ZIF-1 starts to be expressed in a stereotyped pattern of somatic cells after the two-cell stage, starting with the anterior cells (red + blue = purple). Proteins with a ZF1 tag are degraded in somatic cells, starting with the anterior cells (purple to blue). Cells that do not express ZIF-1, such as the posterior germ cell, do not degrade proteins with a ZF1 tag (red). **b** The C-terminal phosphodegrons (CTPD) of OMA-1 are inert until phosphorylated and CTPD-tagged proteins (red) are stable, despite the presence of SCF ubiquitin ligases (blue). Phosphorylation of CTPD occurs during the first mitosis in C. elegans embryos, leading to degradation of CTPD-tagged proteins during the first cell division. **c** Expression of the TIR1 ligase adaptor (blue) is not sufficient to induce robust degradation of proteins tagged with the auxin-inducible degron (AID, red). Addition of auxin family hormones induces degradation of AID-tagged proteins in cells where TIR1 is expressed (purple to blue)

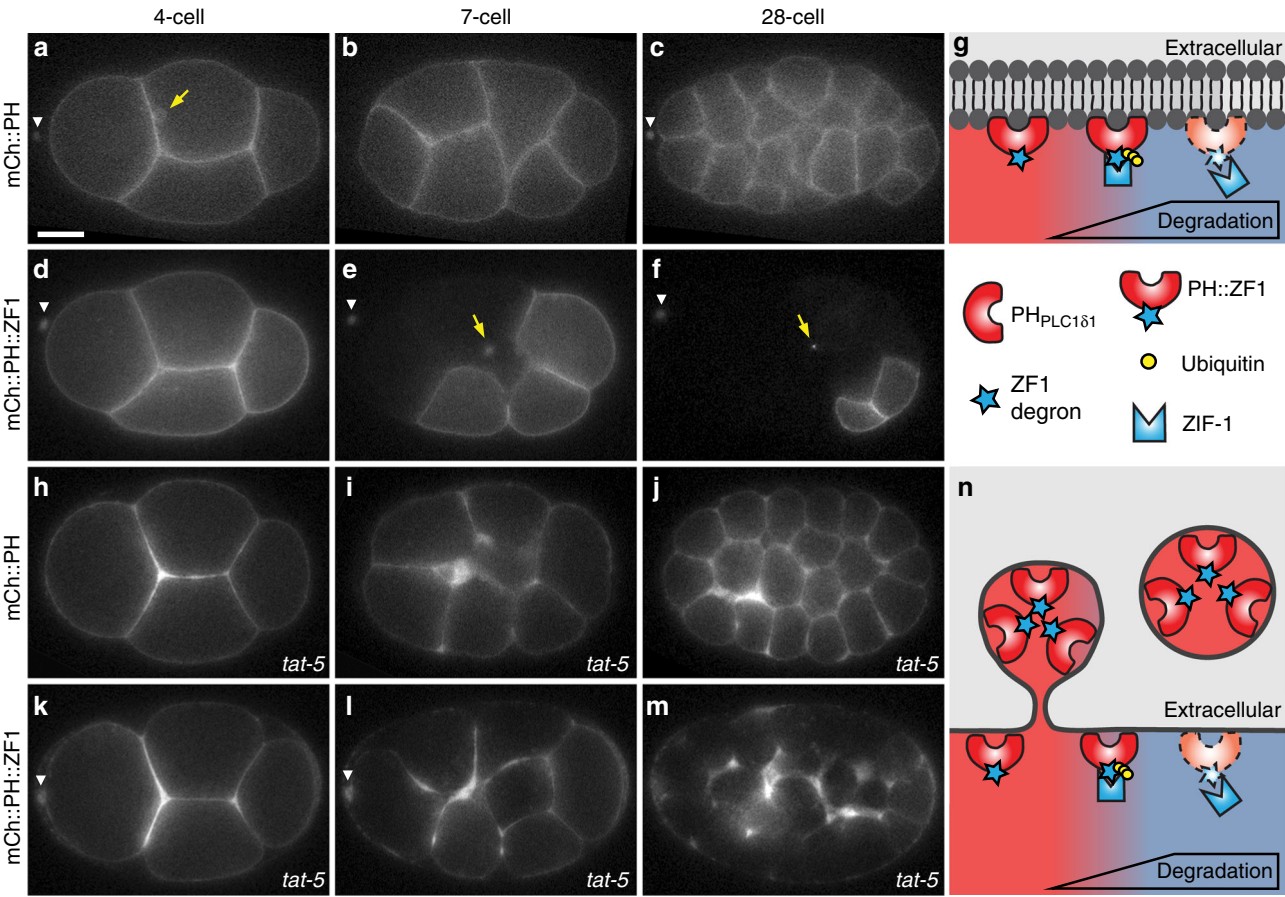

**Fig. 2** The ZF1 degron enables labelling of specific cells and vesicles in *C. elegans* embryos. **a–c** The PH domain of PLC1∂1 binds to lipids in the plasma membrane and an mCherry-tagged $PH_{PLC1∂1}$ reporter (mCh::PH) localizes to the plasma membrane and endocytic vesicles (arrow) in 4-, 7-, and 28-cell embryos ($n = 19$). Scale bar: 10 µm. **d–f** ZIF-1-driven proteasomal degradation of the degron-tagged mCh::PH::ZF1 reporter starts in anterior blastomere (AB) cells during the four-cell stage (**d**), leading to the absence of the mCh::PH::ZF1 fluorescence in anterior AB cells at the seven-cell stage (**e**) and most somatic cells at the 28-cell stage (**f**, $n = 10$). The ubiquitin ligase adaptor ZIF-1 is not expressed in the posterior germ line or in anterior polar bodies (arrowhead), resulting in the persistence of mCh::PH::ZF1 in these cells. Arrows indicate labelled intracellular vesicles, which are protected from proteasomal degradation by intervening membranes. See also Supplementary Movie 1. Anterior is left, dorsal is up. **g** The lipid-binding PH domain tagged with the ZF1 degron is recognized by the ubiquitin ligase adaptor ZIF-1 and ubiquitinated. Polyubiquitination leads to proteasomal degradation of degron-tagged fluorescent reporters (dotted lines). **h–j** Embryos treated with *tat-5* RNAi show increased mCh::PH membrane labelling due to accumulated microvesicles at the 4-, 7-, and 28-cell stage ($n = 11$). **k** Increased mCh::PH::ZF1 is visible at the four-cell stage in embryos treated with *tat-5* RNAi. **l–m** Gradual degradation of mCh::PH::ZF1 in somatic cells facilitates visualization of released microvesicles in a 7- and 28-cell embryo ($n = 22$). Due to degradation of cytosolic mCh::PH::ZF1 reporters at the plasma membrane, even small amounts of released microvesicles are easily visible. See also Supplementary Movie 1 and Supplementary Fig. 2. **n** Degron-tagged PH domain reporters are released in extracellular vesicles that bud from the plasma membrane before ZIF-1 expression. ZIF-1 expression leads to the ubiquitination and proteasomal degradation of cytosolic ZF1-tagged reporters (dotted lines). ZF1-tagged membrane reporters in released vesicles are not ubiquitinated or degraded and maintain fluorescence

## Results

**Degron-tagged reporters are degraded in the cytosol**. To determine whether degron-tagged reporters would be useful for cell biological approaches, we tested whether an endogenous degradation system was capable of degrading abundant reporter proteins and examined whether the increased proteasomal load had negative effects on cells. We tagged a membrane-binding domain, the PH domain of rat PLC1∂1, with the ZF1 degron from *C. elegans* PIE-1 and expressed it in worm embryos. Similar to an mCherry-tagged PH reporter (Fig. 2a), the cytosolic mCh::PH::ZF1 reporter initially localizes to the plasma membrane (Fig. 2d). Thus, the degron tag did not disrupt the normal localization of the reporter.

ZF1-mediated degradation begins in anterior somatic cells at the four-cell stage, due to the onset of expression of the ubiquitin ligase adaptor protein ZIF-1[14]. While mCh::PH fluorescence persists in developing embryos (Fig. 2b, c), mCh::PH::ZF1 is progressively degraded, starting with anterior somatic cells (Fig. 2e–g, Supplementary Movie 1). ZIF-1 is not expressed in the germ lineage, resulting in persistent fluorescence in a couple of posterior cells (Fig. 2e, f). ZIF-1 expression also does not occur in two small cell corpses born at the anterior side of the embryo during meiosis[14,20]. These polar bodies maintain mCh::PH::ZF1 fluorescence (arrowheads in Fig. 2). Thus, the degron tag led to rapid degradation of a highly expressed, exogenous reporter in cells where the ligase adaptor was expressed and the reporter could therefore be ubiquitinated.

As ZIF-1 has a number of known targets whose proteasomal degradation is important for embryonic development[13], we tested whether the expression of ZF1-tagged reporters disrupted development. Stable transgenic strains expressing various ZF1-tagged reporters were fertile and had similar numbers of viable progeny that did not show developmental delays (Supplementary Fig. 1a). In fact, most ZF1-tagged reporter embryos developed

significantly faster than corresponding reporter strains without the degron (Supplementary Fig. 1b) and were less likely to show overexpression defects (Supplementary Fig. 1c–f). This suggests that degron reporters can be tolerated better than other overexpressed reporters and do not generally disrupt development.

**Degron reporters specifically label EVs**. In addition to labelling the plasma membrane, mCh::PH and mCh::PH::ZF1 labelled intracellular vesicles (arrows in Fig. 2), some of which maintained their fluorescence in the mCh::PH::ZF1 strain (Supplementary Movie 1). As mCh::PH::ZF1 on the cytosolic face of vesicles would be accessible for ubiquitination and proteasomal degradation, the persistence of the degron reporter suggests that it is protected from degradation by intervening membranes. We hypothesized that the PH::ZF1 reporter persisted in EVs or other cell debris that are taken up by the cell during endocytosis.

To test whether the degron-tagged PH reporter could be used to specifically label and track EVs released in vivo, we examined microvesicles. Microvesicles are 90–500 nm vesicles that arise from plasma membrane budding, aka ectocytosis[8,21]. In wild-type embryos, microvesicles are difficult to detect due to their low abundance and proximity to the plasma membrane. Microvesicle budding is normally inhibited by the TAT-5 lipid flippase, resulting in continuous microvesicle release when *tat-5* is knocked down, as demonstrated by electron tomography[21]. In mCh::PH embryos, microvesicle overproduction is visible as thickened membrane labelling between *tat-5* knockdown cells (Fig. 2h–j) in comparison to control embryos (Fig. 2a–c). However, small patches of microvesicles are difficult to detect over the background of the plasma membrane fluorescence. In contrast, released microvesicles are clearly visible after *tat-5* knockdown using the mCh::PH::ZF1 reporter (Fig. 2l, m)[2], due to proteasomal degradation of the plasma membrane label (Fig. 2n). Released EVs are then visible floating between the embryo and the eggshell (Supplementary Movie 1). Thus, degron tagging a general plasma membrane reporter reveals microvesicles and their movement in vivo.

To determine whether this was a specific feature of the ZF1 degron or the ECS ubiquitin ligase, we tested whether another degron could be used with SCF ubiquitin ligases to label EVs. We tagged the PH reporter with a C-terminal fragment of OMA-1 (aa 219–378) containing two phosphorylation sites important for degradation[16], which we named the C-terminal phosphodegrons (CTPD) (Fig. 3d). Early during the one-cell stage, mCh::PH::CTPD localized brightly to the plasma membrane, but began to be degraded during the first mitotic division (Fig. 3a, Supplementary Movie 2). Degradation was transient and some mCh::PH::CTPD persisted on the plasma membrane after the two-cell stage (Fig. 3c). Even with partial degradation, microvesicles could readily be observed with mCh::PH::CTPD after *tat-5* RNAi treatment (Fig. 3b, Supplementary Movie 2). Thus, even a partial loss of plasma membrane signal enhanced visualization of EVs.

We also tested whether it was possible to label EVs by degron-tagging transmembrane proteins. In contrast to the proteasomal degradation of cytosolic proteins, ubiquitination of post-Golgi transmembrane proteins leads to degradation within lysosomes[12]. The syntaxin SYX-4 is a single-pass transmembrane protein that localizes to the plasma membrane and endocytic vesicles (Supplementary Fig. 2a–c)[22]. We tagged the cytosolic domain of SYX-4 with the ZF1 degron to make it accessible to ZIF-1. Degron-tagged GFP::ZF1::SYX-4 localizes normally before the onset of ZIF-1 expression (Supplementary Fig. 2d), after which GFP::ZF1::SYX-4 accumulates in intracellular vesicles and is lost from the plasma membrane (Supplementary Fig. 2e). These

vesicles eventually disappear from ZIF-1-expressing cells (Supplementary Fig. 2f), consistent with ubiquitin-driven endocytosis, sequestration inside intraluminal vesicles, and lysosomal degradation (Supplementary Fig. 2i). To test whether lysosomal degradation of a transmembrane protein can label EVs, we treated the GFP::ZF1::SYX-4 reporter strain with *tat-5* RNAi to induce microvesicle release. Similar to the degron-tagged PH reporter (Fig. 2k–n), GFP::ZF1::SYX-4 accumulates around cells after *tat-5* knockdown (Supplementary Fig. 2g, h)[21]. Thus, both membrane-associated and transmembrane proteins can be tagged with degrons to specifically label EVs.

**Degron protection assay reveals topology of tagged protein**. We next tested whether degron tagging could reveal insights into protein topology. Clathrin is enriched at the cell surface after *tat-5* knockdown[21], but it was unclear whether this was due to increased clathrin inside the plasma membrane or due to the release of clathrin in EVs that accumulated next to the plasma membrane (Fig. 4f). Both possibilities were plausible, as clathrin-binding proteins are increased at the plasma membrane after *tat-5* knockdown, including ESCRT proteins[21], and as clathrin was found in purified *Drosophila* and mammalian EVs[23,24]. Therefore, we asked whether a degron-tagged clathrin heavy chain reporter would be protected from degradation inside EVs or be accessible to degradation at the cell cortex. Consistent with previous results[21], we initially saw increased ZF1::mCh::CHC-1 and GFP::ZF1::PH labelling clathrin and membrane at cell contacts in embryos treated with *tat-5* RNAi (Fig. 4b) compared to control embryos (Fig. 4a). This increase persisted in posterior *tat-5* knockdown cells (Fig. 4d, e), but after ZIF-1 expression began in anterior cells, ZF1::mCh::CHC-1 disappeared from the cell surface in both control and *tat-5* RNAi-treated embryos (Fig. 4c–e), while GFP::ZF1::PH persisted in EVs after *tat-5* knockdown (Fig. 4d). This demonstrates that the increased clathrin signal is due to association with the plasma membrane and not due to clathrin trapped within EVs. Thus, degron-tagged reporters reveal whether a protein is inside or outside the plasma membrane.

**Degron reporters enable tracking of phagocytosed cargo**. To test whether the specific labelling by degron tags facilitates long-term tracking, we observed the two polar bodies in which ZF1 degradation does not occur (Fig. 2d–f). As polar bodies are dying cells, they have a nucleus and can be tagged with chromosome reporters like histone H2B, in addition to the PH domain[20]. Both polar bodies are initially found on the anterior surface of the embryo (Fig. 5a, d). The first polar body is trapped in the eggshell[25], while the second polar body (2PB) is phagocytosed by an anterior cell[20]. Because mCh::H2B labels all nuclei in the embryo, it can be challenging to track the 2PB phagosome among the many dividing nuclei (Fig. 5b, c, Supplementary Movie 3). In contrast, the degron-tagged ZF1::mCh::H2B reporter disappears sequentially from somatic nuclei, leaving the two polar bodies as the only fluorescent structures on the anterior half of the embryo (Fig. 5e, Supplementary Movie 3). This confirms that the intervening cell corpse and phagosome membranes protect ZF1-tagged proteins from proteasomal degradation (Fig. 5g). Improving the signal-to-noise ratio with degron-tagged reporters also improves automated tracking of the 2PB by removing overlapping traces (Fig. 5c, f). Tubulation of the 2PB phagosome into small vesicles can also be followed with degron-tagged reporters (Supplementary Movie 1, left)[20]. Thus, degron-tagged reporters reveal organelle dynamics and facilitate tracking by removing background labelling.

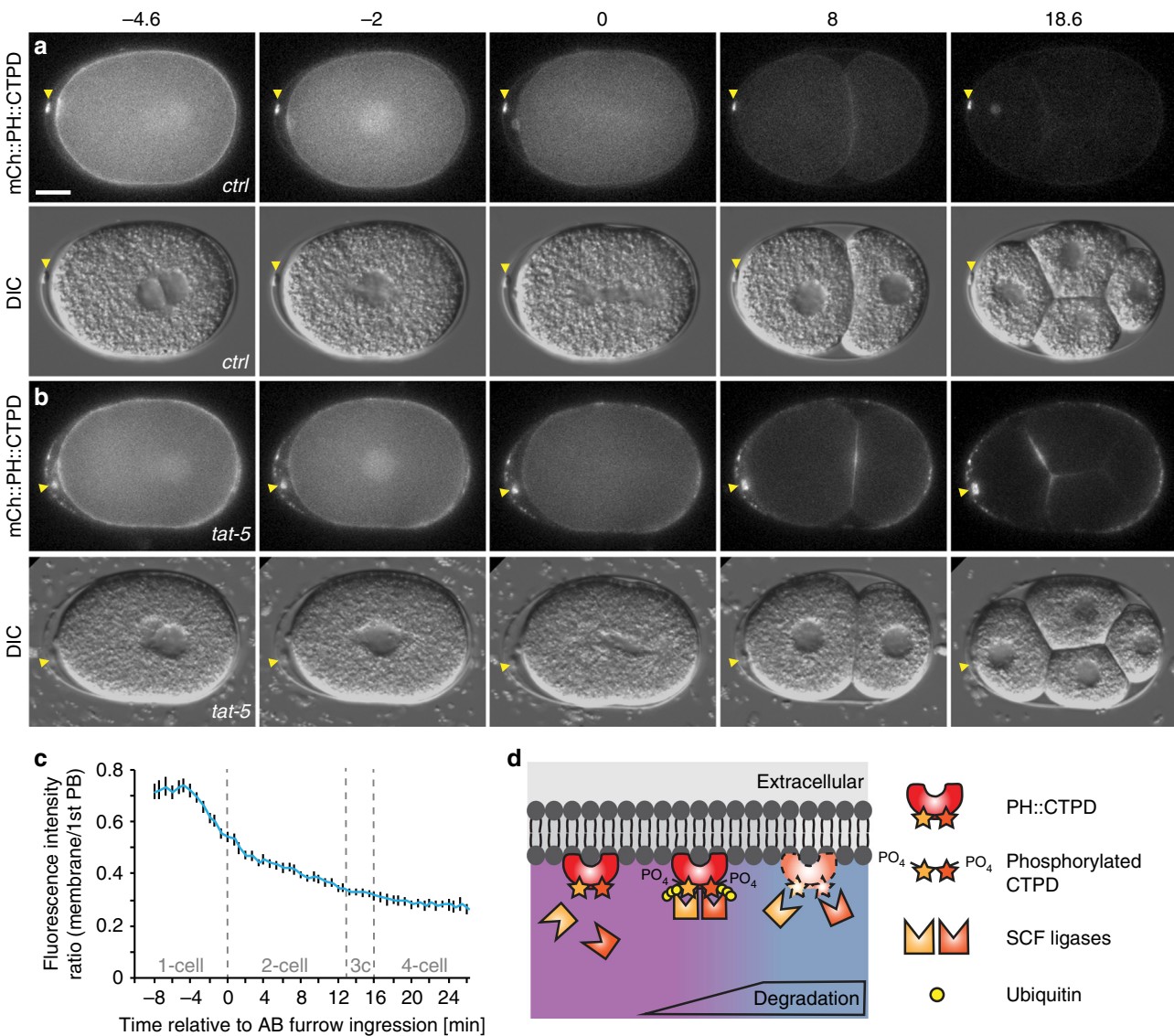

**Fig. 3** The CTPD domain leads to transient degradation and labelling of extracellular vesicles. **a** An mCherry and CTPD-tagged PH reporter localizes to the plasma membrane in one-cell embryos, but begins to be degraded during mitosis. Weaker fluorescence persists in embryonic cells from the two-cell stage on, but polar bodies remain brightly labelled (arrowhead, *n* = 42). Scale bar: 10 μm. **b** After *tat-5* knockdown, the mCh::PH::CTPD reporter still degrades, but bright patches of extracellular vesicles are visible in the eggshell, floating around the embryo, and on the embryo surface, in addition to the brightly labelled polar bodies (arrowhead, *n* = 43). See also Supplementary Movie 2. **c** Loss of mCh::PH::CTPD fluorescence from the cell surface begins ~3 min before cytokinetic furrow ingression, but degradation tapers off leaving residual mCh::PH::CTPD fluorescence after the two-cell stage. Fluorescence intensity was normalized to the first polar body to correct for photobleaching. Bars represent mean ± s.e.m. (*n* = 5–12). Source data are provided as a Source Data file. **d** Phosphorylation of the CTPD degrons leads to recognition of the CTPD by multiple SCF ubiquitin ligases, ubiquitination, and degradation

**Degron reporters reveal membrane topology and dynamics.** Degron-tagged reporters also allow the specific labelling of a single membrane or surface of a membrane. The close proximity of the corpse plasma membrane and the engulfing phagosome membrane make it difficult to distinguish their signals using fluorescence microscopy[26,27], unless cells have distinct transcriptional programmes, i.e. neuronal corpses engulfed by non-neuronal cells. Using degron reporters, this can be achieved even with sister cells in vivo by degrading the reporter localized to the phagosome membrane, leaving the corpse membrane preferentially labelled (Fig. 6d). For example, as polar bodies do not enter mitosis, their plasma membranes remain brightly labelled after mCh::PH::CTPD degradation occurs in the embryo (Fig. 6a). After phagocytosis, the 2PB membrane appears as a bright hollow sphere in a weakly labelled cell using the mCh::PH::CTPD reporter (Fig. 6b). In order to degrade or recycle corpse contents, the plasma membrane of engulfed corpses must be disrupted within the safety of the phagosome membrane[28]. 2PB membrane breakdown can be visualized using the mCh::PH::CTPD reporter, which is seen dispersing throughout the phagosome lumen (Fig. 6c). Similar results were found using the mCh::PH::ZF1 reporter (Supplementary Movie 1)[20]. Thus, by depleting fluorescent reporters from membrane surfaces facing the cytosol, degron tagging enables examination of specific membranes and their dynamics.

We next asked whether degron-tagged reporters can be used to assess nuclear topology. Other components of the ECS ubiquitin ligase complex are found in both the cytosol and nucleus, but the ZIF-1 ligase adaptor appears cytosolic[29]. To test whether ZIF-1 degradation was restricted to the cytosol (Fig. 7a), we tagged the

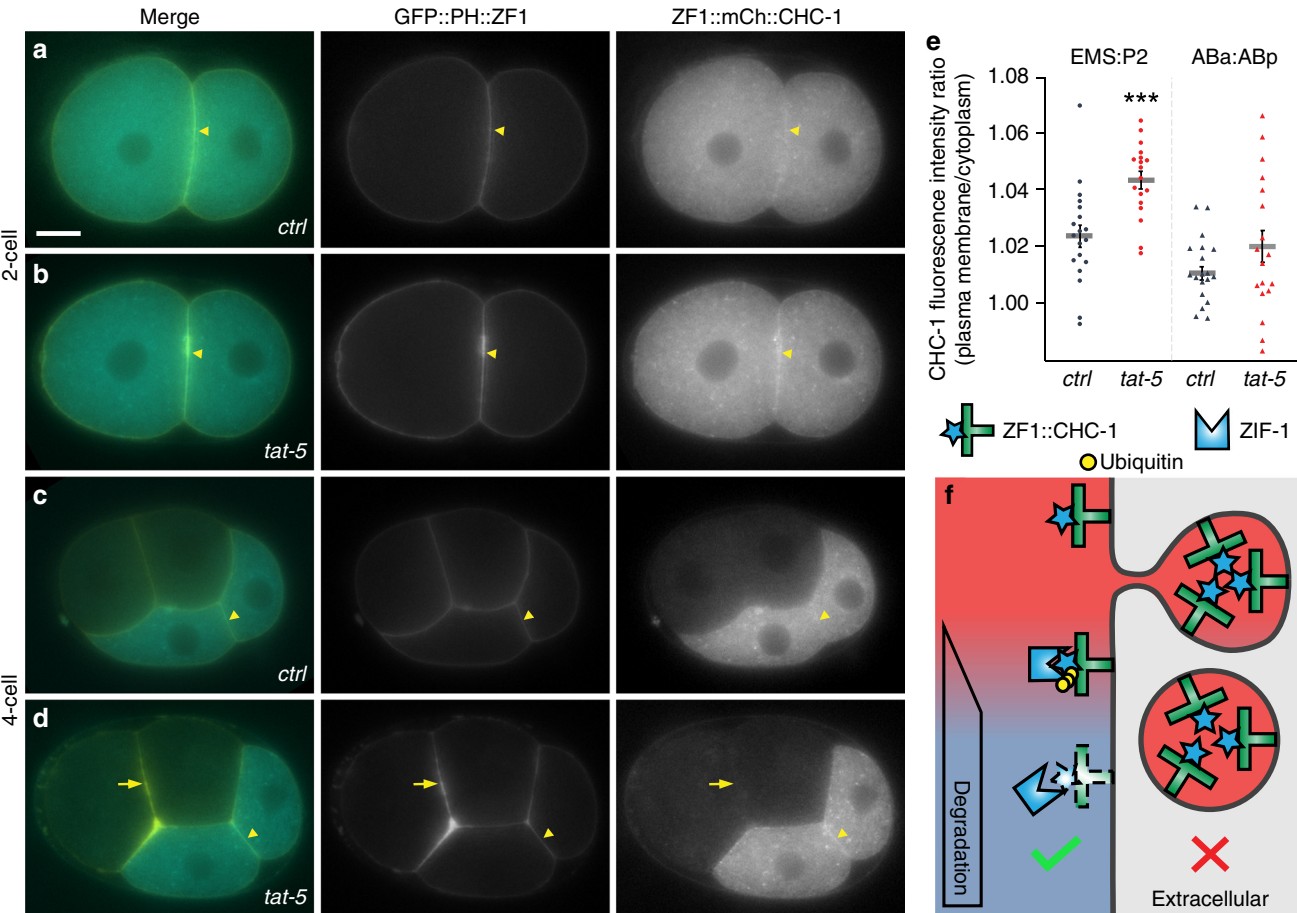

**Fig. 4** Degron protection assay reveals protein topology. **a** A degron-tagged clathrin reporter ZF1::mCh::CHC-1 initially localizes to the plasma membrane (arrowhead) and intracellular puncta (*n* = 9). Scale bar: 10 μm. **b** After *tat-5* knockdown, ZF1::mCh::CHC-1 is enriched at the plasma membrane (arrowhead, *n* = 14). **c** After expression of ZIF-1 begins in anterior cells, ZF1::mCh::CHC-1 is degraded throughout the anterior cells in control embryos (*n* = 19). **d** Although ZF1::mCh::CHC-1 is still enriched at the plasma membrane in posterior cells (arrowhead) in *tat-5* RNAi-treated embryos, ZF1::mCh::CHC-1 is lost from the plasma membrane in anterior cells (arrow, *n* = 18), indicating that clathrin is accessible to ubiquitin ligases. GFP::PH::ZF1 labels the plasma membrane and extracellular vesicles. **e** Quantification of clathrin enrichment on a posterior cell contact (EMS:P2) or anterior cell contact (ABa:ABp) compared to the neighbouring cytoplasm at the four- and six-cell stage from two independent experiments. ZF1::mCh::CHC-1 fluorescence was significantly increased at the posterior EMS:P2 cell contact (\*\*\**p* < 0.001 using Student's *t*-test, *ctrl n* = 19, *tat-5* RNAi *n* = 18). No change was observed at anterior cell contacts (*p* > 0.05). Bars represent mean ± s.e.m. Source data are provided as a Source Data file. **f** If clathrin were in extracellular vesicles, ZF1::mCh::CHC-1 would be protected from ZIF-1-mediated degradation. However, as clathrin is inside the plasma membrane, ZF1::mCh::CHC-1 is accessible to ZIF-1-mediated degradation

nuclear lamin LMN-1 with the ZF1 degron to examine the dynamics of the nuclear cortex during cell division[30]. Prior to the onset of ZIF-1 expression, the fluorescence intensity of the mKate2::ZF1::LMN-1 reporter is comparable between the anterior and posterior nuclei (Fig. 7c). After ZIF-1 expression begins in the two anterior daughter cells, the interphase levels of mKate2::ZF1::LMN-1 gradually drop in comparison to posterior cells (Fig. 7d, g), suggesting that some degradation occurs despite the intact nuclear envelope. However, the degradation of mKate2::ZF1::LMN-1 occurred faster during mitosis (Fig. 7e–g, Supplementary Movie 4). The nuclear envelope breaks down during mitosis for chromosome segregation[31], which would allow cytosolic ZIF-1 to target ZF1-tagged nuclear proteins (Fig. 7b).

To confirm that loss of fluorescence was due to ZIF-1-mediated degradation and not due to morphological changes in the nuclear envelope during the cell cycle, we treated reporter embryos with *zif-1* RNAi to inhibit degradation. The fluorescence of mKate2::ZF1::LMN-1 fluctuated as cells divided (Fig. 7g), but fluorescence persisted in anterior cells (Fig. 7i). We normalized the control curve to the *zif-1* curve to remove

changes due to nuclear morphology, revealing 4–6 times faster degradation during NEBD (Fig. 7h). Thus, although a pool of mKate2::ZF1::LMN-1 is accessible to ZIF-1 during interphase, mKate2::ZF1::LMN-1 is largely protected from proteasomal degradation by the nuclear envelope (Fig. 7a). After treating mKate2::ZF1::LMN-1 embryos with *zif-1* RNAi, we noticed an increase in nuclear morphology defects (Supplementary Fig. 1c). However, the mKate2::ZF1::LMN-1 reporter did not result in other defects typical of LMN-1 overexpression, even when *zif-1* was knocked down (Supplementary Fig. 1d–f). Therefore, degron tagging provides probes for membrane topology that can investigate the dynamics of NEBD.

As LMN-1 is largely immobile in the nuclear lamina[32], the observation that some degradation occurred during the interphase raised the possibility that ubiquitination of the degron-tagged reporter was altering LMN-1 dynamics. To determine whether mKate2::ZF1::LMN-1 became mobile after ZIF-1 expression, we performed fluorescence recovery after photobleaching experiments in an anterior daughter nucleus (Supplementary Fig. 3a). There was no significant recovery of mKate2::ZF1::LMN-

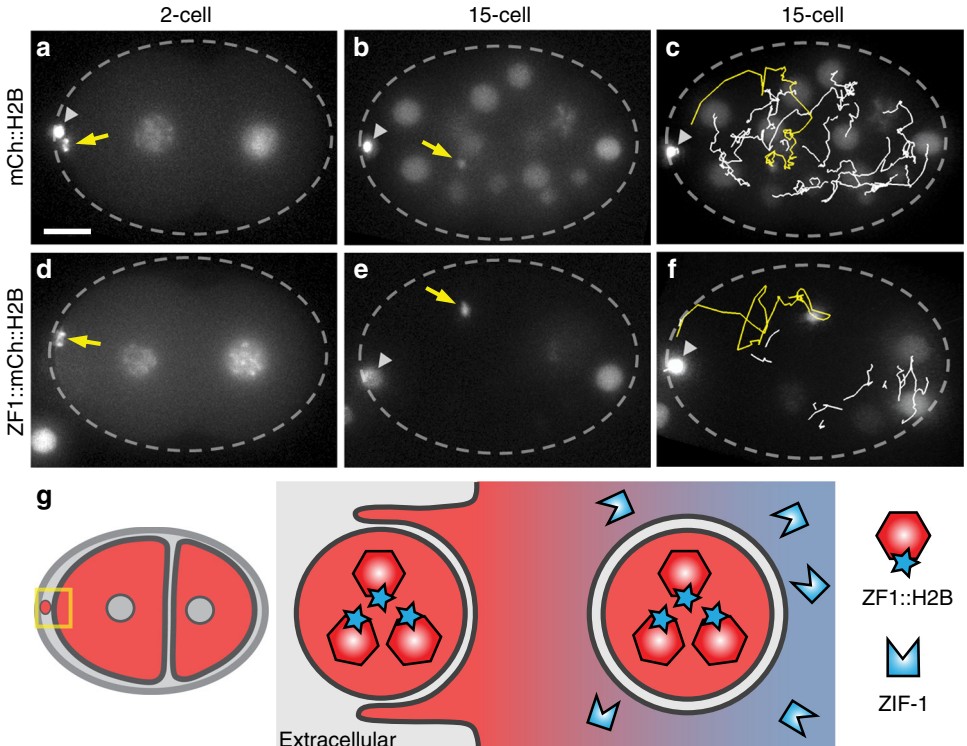

**Fig. 5** Degron reporters label phagocytosed cell debris and enable tracking. **a–c** The histone reporter mCh::H2B labels chromosomes in embryonic nuclei and two polar bodies ($n = 13$). The first polar body (arrowhead) is trapped in the eggshell (dashed oval). Scale bar: 10 μm. **a** The second polar body (2PB, arrow) neighbours the anterior blastomere (AB cell) in a two-cell embryo. **b** The 2PB is engulfed in a phagosome in a 15-cell embryo. Due to H2B fluorescence from surrounding embryonic nuclei, it is hard to track the 2PB. **c** Automated tracking of the 2PB (yellow) results in crossing tracks from nearby nuclei (white), increasing the potential need for manual correction. **d–f** Embryos labelled with ZF1-tagged mCh::H2B ($n = 12$). **d** The released 2PB neighbours a two-cell embryo. **e** After engulfment, the 2PB is easily trackable due to degradation of ZF1::mCh::H2B in somatic cell nuclei. See also Supplementary Movie 3. **f** Automated tracking of the 2PB (yellow) is simplified by removing the label of nearby nuclei (white). **g** Degron reporters released in cells or other debris prior to expression of the ZIF-1 ubiquitin ligase adaptor are protected from ZF1-mediated degradation. After engulfment in a second layer of membrane, they are still protected from proteasomal degradation. As degron reporters are degraded in the cytosol, only the reporters within the phagosome remain fluorescent, improving the signal-to-noise ratio

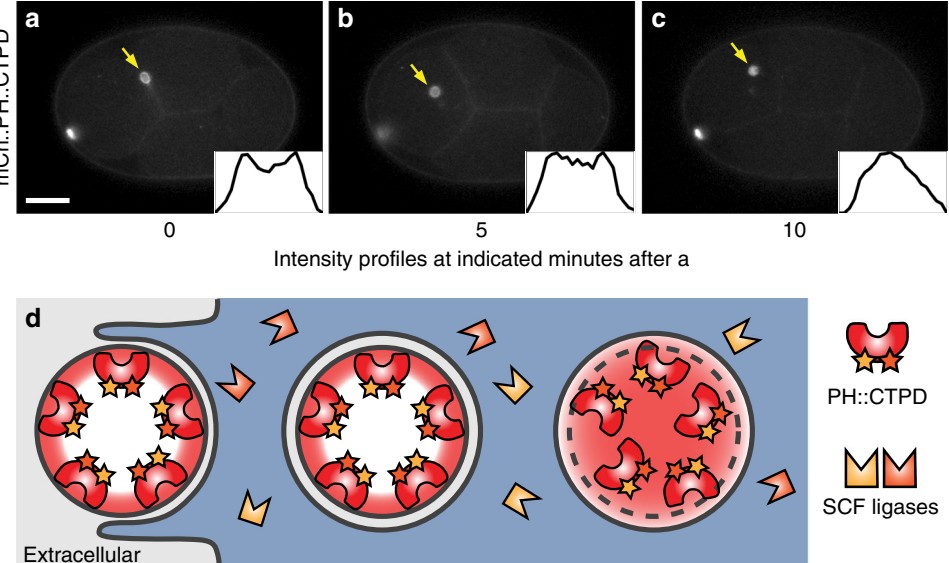

**Fig. 6** Degron reporters label specific membranes. **a** The mCh::PH::CTPD reporter brightly labels the plasma membrane of the polar bodies and weakly labels the plasma membrane of embryonic cells at the four-cell stage ($n = 22$). Scale bar: 10 μm. **b** After phagocytosis of the second polar body (2PB, arrow), the plasma membrane of the 2PB is distinctly visible. **c** Several minutes later, the corpse plasma membrane breaks down inside the phagolysosome and the mCh::PH::CTPD reporter disperses throughout the phagolysosome lumen, as demonstrated by line scans through the phagolysosome (insets). See also Supplementary Movie 2. **d** The mCh::PH::CTPD reporters inside the corpse membrane are protected from degradation before and after phagocytosis. Model of membrane breakdown inside the phagolysosome, demonstrating the shape change from a hollow to a filled sphere

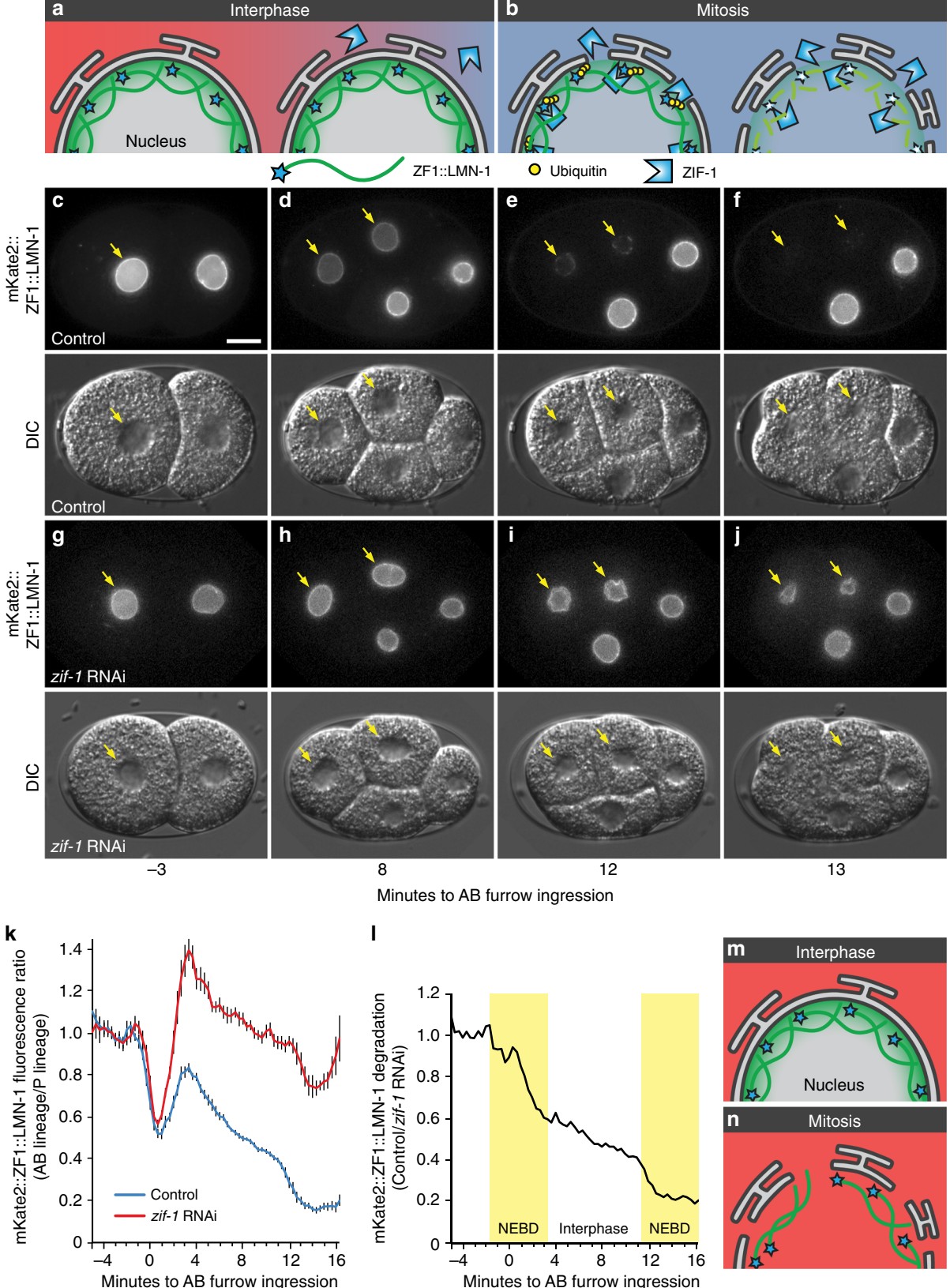

1 fluorescence during the interphase (Supplementary Fig. 3b), suggesting that the mobility of LMN-1 was unchanged by ubiquitination. Thus, quantification of degron-tagged reporters can be a tool to examine protein dynamics in addition to organelle dynamics.

**Ubiquitin ligase adaptor localization determines degradation**. We next tested whether protection by internal membranes was a specific feature of ZIF-1 and ECS ubiquitin ligases or whether the nuclear envelope could also protect targets of SCF ubiquitin ligases from degradation. As SCF ubiquitin ligase complexes are

**Fig. 7** Degron reporters reveal nuclear envelope topology during cell division. **a** During interphase, the nuclear envelope is intact, hindering ZIF-1 binding to ZF1-tagged reporters inside the nuclear envelope. **b** During mitosis, the nuclear envelope is remodelled into the endoplasmic reticulum, allowing binding of ZIF-1 to ZF1-tagged reporters, which leads to ubiquitination and rapid proteasomal degradation. **c** Prior to ZIF-1 expression, mKate::ZIF1::LMN-1 is similarly bright in the nuclear matrix of the anterior AB cell (arrow) and the posterior cell ($n = 14$). Scale bar: 10 μm. **d** After the onset of ZIF-1 expression in anterior daughter cells (ABx), mKate2::ZIF1::LMN-1 starts to lose fluorescence in anterior cells (arrows). **e, f** During mitosis, the nuclear envelope is disassembled, leading to rapid proteasomal degradation of mKate2::ZIF1::LMN-1. DIC images show nuclear morphology during cell cycle progression. **g–j** mKate2::ZIF1:: LMN-1 is maintained in anterior cells after *zif-1* knockdown ($n = 11$). DIC images show nuclear morphology during cell cycle progression. See also Supplementary Movie 4. **k** Quantification of mKate2::ZIF1::LMN-1 fluorescence in the dorsal ABp nucleus compared to the ventral EMS nucleus with ($n = 5$–11) and without *zif-1* RNAi treatment ($n = 5$–14). Bars represent mean ± s.e.m. **l** Normalizing mKate2::ZIF1::LMN-1 to *zif-1* RNAi-treated embryos demonstrates that fluorescence drops 4–6 times faster during nuclear envelope breakdown (NEBD) in anterior cells (yellow boxes, AB: 16% per min, ABx: 11% per min), when ZIF-1 has full access to mKate2::ZIF1::LMN-1. Slow decline in the intensity of mKate2::ZIF1::LMN-1 is also visible during interphase (3% per min). NEBD was defined as the first time point the nuclear lamina collapsed in the mother cell until the lamina became round again in daughter cells. Time is given in relation to AB furrow ingression. Source data are provided as a Source Data file. **m, n** In the absence of ZIF-1, ZF1-tagged reporters are stably expressed, because they are not ubiquitinated or degraded during interphase or mitosis

---

found in the nucleus and cytosol, we modified the TIR1 ligase adaptor from plants with a nuclear export signal (NES) and expressed NES-TIR1 in HeLa cells that also expressed a lamin A reporter tagged with a minimized AID (mAID) degron[33]. During interphase, little Venus-mAID-LMNA fluorescence was lost when TIR1 was restricted to the cytosol (Fig. 8a, b, g, Supplementary Movie 5). After cells entered mitosis and NEBD occurred, NES-TIR1 caused rapid degradation of Venus-mAID-LMNA (Fig. 8c, d, g, Supplementary Movie 6). Thus, degron protection assays can also be used in mammalian cells to probe membrane topology.

To confirm that protection from degradation was due to the localization of the ligase adaptor and not due to a protective effect of the nuclear lamina, we co-expressed NLS-TIR1, which has a nuclear localization signal (NLS), with NES-TIR1 (ref. [33]). When TIR1 was localized to both the nucleus and the cytosol, Venus-mAID-LMNA was rapidly degraded during interphase (Fig. 8e, f, h, Supplementary Movie 7). The velocity of interphase degradation by NLS-TIR1 was not significantly different from mitotic degradation by NES-TIR1 (Fig. 8i), indicating that LMNA was not protected from ubiquitination by integration into the nuclear lamina. These results demonstrate that the localization of the ligase adaptor can determine the localization of degradation, providing a strategy to target one pool of a reporter protein for degradation.

**Degron reporters reveal membrane topology during abscission.** In order to understand how degrons in restricted spaces can be ubiquitinated and degraded, we applied degron-tagged reporters to the process of abscission. During cell division, the actomyosin furrow closes around the spindle midbody to form a narrow intercellular bridge[34]. Both sides of the bridge are cleaved during abscission to release a ~1 μm EV called the midbody remnant, which is later phagocytosed (Fig. 9h)[35]. The intercellular bridge no longer permits diffusion between cells ~4 min after furrow ingression[34], but the first cut for abscission does not occur until ~10 min after furrow ingression[36]. Therefore, we asked when actomyosin accumulated in the intercellular bridge was accessible for proteasomal degradation. We tagged non-muscle myosin (NMY-2) with the ZF1 degron and measured the fluorescence intensity of NMY-2::GFP::ZF1 in the bridge between the anterior daughter cells (Fig. 9a)[35]. Degradation of cytoplasmic NMY-2:: GFP::ZF1 was first visible $8 \pm 1$ min after furrow ingression. NMY-2::GFP::ZF1 in the bridge showed a small but significant decline for the next 2 min (Fig. 9g), suggesting that NMY-2 is normally able to diffuse out of the bridge up to 10 min after furrow ingression. Subsequently, NMY-2::GFP::ZF1 in the bridge was protected from proteasomal degradation (Fig. 9b, g), suggesting that either a diffusion barrier had formed or abscission had occurred. Thus, using degron reporters and light microscopy

on living embryos, we could confirm the timing of abscission estimated from electron microscopy data from fixed embryos.

To test whether degron-tagged reporters were able to detect novel phenotypes in abscission mutants, we depleted proteins implicated in abscission, including the ESCRT-I subunit TSG-101 and the septin UNC-59. At the onset of ZIF-1-mediated degradation, NMY-2::GFP::ZF1 labelled intercellular bridges in *tsg-101* or *unc-59* mutants normally (Fig. 9c, e). In contrast to control embryos where NMY-2::GFP::ZF1 fluorescence persisted through midbody release and phagocytosis (Fig. 9b), NMY-2:: GFP::ZF1 fluorescence intensity continued to drop significantly in the bridge of both *tsg-101* and *unc-59* mutants (Fig. 9d, f–g), and this drop was dependent on ZIF-1 expression[35]. Phagocytosis of the midbody remnant was also delayed in both *tsg-101* and *unc-59* mutants (Fig. 9d, f)[34,35], consistent with a delay in abscission. These findings demonstrate that NMY-2 is able to be degraded when abscission is delayed (Fig. 9i). No defect was detected for *tsg-101* knockdown using a dextran diffusion assay[34], demonstrating the high sensitivity of the degron protection assay for detecting abscission defects.

**Single membrane organelles protect degrons from degradation.** As all of the model systems we examined involved two membrane bilayers between the ubiquitin ligase and the degron-tagged reporter (nucleus, EV, phagocytosed cell debris), we tested whether a single membrane bilayer would protect a reporter from degradation. We expressed a degron-tagged reporter in the secretory pathway and maintained it in the endoplasmic reticulum (ER) using a C-terminal KDEL sequence[37] (Fig. 10d). The ss:: TagRFP-T::ZF1::KDEL reporter localized to the ER, similar to established reporters (Fig. 10a, b). After ZIF-1 expression began in anterior cells, ss::TagRFP-T::ZF1::KDEL persisted with no measurable loss of fluorescence (Fig. 10b, c). Thus, a single membrane bilayer protects degron-tagged reporters from degradation, confirming that degron protection assays can be applied to intracellular organelles.

**Degron-tagged reporters do not degrade binding partners.** Ligase adaptors can target proteins for ubiquitination through intermediary binding partners, including nanobodies[33], which raised the possibility that degron-tagged reporters would lead to the degradation of untagged binding partners. This could be especially relevant when the reporter protein assembles into larger complexes, such as histones or the nuclear lamina, or dimerizes, such as NMY-2. To test whether degron-tagged proteins lead to the degradation of untagged proteins, we generated strains expressing fluorescent reporter proteins for H2B, LMN-1, and NMY-2 both with and without the ZF1 degron tag. We

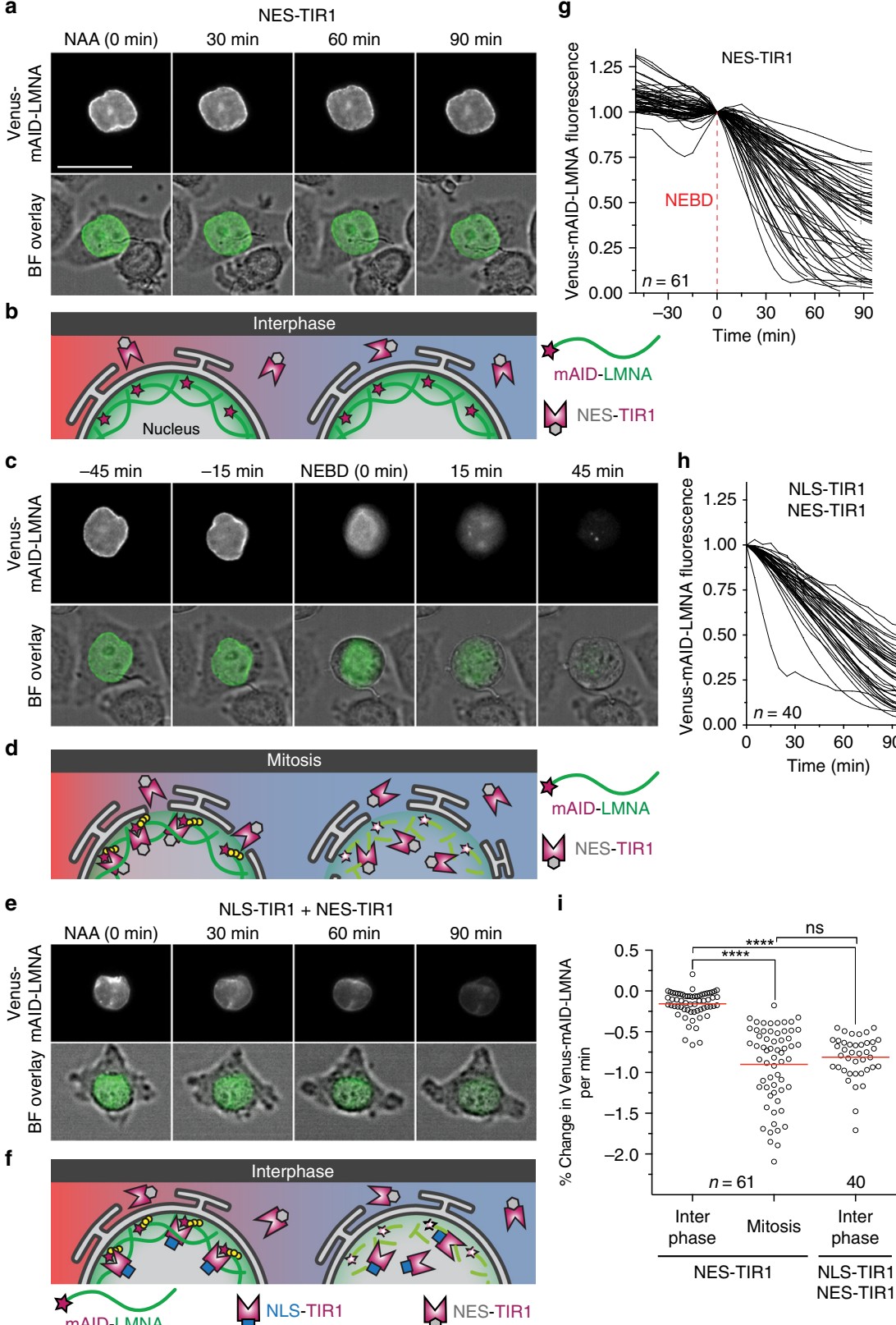

measured the intensity of fluorescence for each reporter and found that although the ZF1-tagged reporter degraded, the reporter without the degron did not show significant changes in comparison to a strain that did not express any degron-tagged reporter (Supplementary Fig. 4). Thus, degron tagging does not generally lead to degradation of protein complexes.

## Discussion

In summary, degron-mediated degradation is more than a loss-of-function technique; it is a powerful tool to study dynamics from the level of proteins to organelles to cells. By removing cytoplasmic fluorescence, degron tags improve the visibility of extracellular, luminal, or nuclear reporters and enable long-term

**Fig. 8** Localization of the ubiquitin ligase adaptor spatially controls degradation. **a** Representative time-lapse images of a HeLa cell transiently expressing a nuclear export signal-tagged ubiquitin ligase adaptor (NES-TIR1) and a Venus- and mAID-tagged lamin LMNA. 0.5 mM NAA was added at $t = 0$ to induce auxin-dependent degradation. During interphase, Venus-mAID-LMNA in the nuclear lamina is protected from ubiquitination by cytosolic NES-TIR1 and fluorescence persists in the presence of NAA. Scale bar: 30 μm. See also Supplementary Movie 5. **b** Model of mAID-LMNA protection from cytosolic NES-TIR1 during interphase. **c** The same cell is shown entering mitosis, which is visible by cell rounding in the brightfield (BF) overlay. After nuclear envelope breakdown (NEBD, $t = 0$), Venus-mAID-LMNA is accessible by cytosolic NES-TIR1 and fluorescence rapidly disappears. See also Supplementary Movie 6. **d** mAID-tagged reporters in the nucleus are accessible to cytosolic NES-TIR1 after NEBD occurs during mitosis. **e** Interphase HeLa cell stably expressing a nuclear localization signal-tagged TIR1 (NLS-TIR1) and NES-TIR1 from a single mRNA in addition to Venus-mAID-LMNA. After treatment with 0.5 mM NAA ($t = 0$), Venus-mAID-LMNA fluorescence rapidly disappears during interphase. See also Supplementary Movie 7. **f** Model of mAID-LMNA accessibility to nuclear NLS-TIR1 during interphase. **g** Quantification of Venus-mAID-LMNA fluorescence before and after NEBD ($t = 0$) in the presence of cytosolic NES-TIR1 and NAA. Values were normalized to NEBD. **h** Quantification of Venus-mAID-LMNA fluorescence in interphase cells in the presence of cytosolic NES-TIR1, nuclear NLS-TIR1, and NAA. Values were normalized to $t = 0$ (NAA addition). **i** Comparison of Venus-mAID-LMNA degradation velocities from single cells. The degradation velocity was determined for 45 min before NEBD (interphase) and 45 min after NEBD (mitosis) for NES-TIR1-expressing cells from three independent experiments or during the first 45 min after NAA addition for cells expressing both NLS-TIR1 and NES-TIR1 from two independent experiments. Bars indicate the mean of the indicated number of cells. Significance according to unpaired multi-comparison Kruskal–Wallis test with Dunn's statistical hypothesis testing (****$p < 0.0001$; ns, $p > 0.9999$). Source data are provided as a Source Data file

tracking. Degron-tagged reporters reveal insights on an epifluorescence microscope that are typically limited to super-resolution or electron microscopy on fixed samples. Our studies have focused on structures that are protected by membrane bilayers, but this approach should work for any structure resistant to ubiquitination or diffusion. Thus, degron tagging is an important addition to the cell biologist's toolbox.

As degron tags are widely used in cell extracts, cell culture, and in vivo, this approach can visualize structures in many systems. We started with endogenous degrons in *C. elegans* embryos for their simplicity, only requiring expression of a degron-tagged reporter. Heterologous expression of ZIF-1 in worms or zebrafish can also degrade ZF1-tagged proteins and could be adapted to more systems[15,38]. Fusing an anti-GFP nanobody to endogenous ubiquitin ligase adaptors like ZIF-1 enables spatial control of degradation of GFP-tagged proteins in *C. elegans*, *Drosophila*, plants, and zebrafish[38–41], which allows existing GFP-tagged reporters to be used for degron protection assays. Heterologous expression of the AID and TIR1 ligase adaptor is also used in various animal models[18,33,42]. We used AID in mammalian cells to demonstrate that altering the localization of the ubiquitin ligase adaptor is sufficient to target degradation to the nucleoplasm or cytoplasm, consistent with observations on endogenous ligase adaptors[43]. Thus, the spatial control of degradation enabled by heterologous expression of ligase adaptors refers not only to expression in specific cell types, but also to specific organelles. As different ubiquitin ligases are found in different cellular compartments[44], the choice of degron–ligase pair can enable degron protection assays in a range of organelles. For example, the Ab-SPOP/FP system uses a BCR ubiquitin ligase complex including Cul3 to specifically target GFP-tagged proteins in the nucleus for degradation[38,45], but Cul3 also localizes to the Golgi. Furthermore, ubiquitin-mediated degradation can be regulated by small molecule drugs, temperature, or light[11], offering many modalities to control the timing and localization of protein degradation. Thus, degron protection assays can readily probe cell biology in many model systems.

To avoid loss-of-function effects from induced degradation of degron-tagged proteins, our studies were performed with isolated protein domains or in the background of the untagged endogenous protein. Still, degradation of an overexpressed degron-tagged protein could alter some processes. We found that strains expressing degron-tagged fluorescent reporters were healthier than strains expressing fluorescent-tagged reporters. In fact, we only observed effects on nuclear morphology after inhibiting degron-mediated degradation of a LMN-1 reporter (Supplementary Fig. 1c). Regardless, degrons like CTPD or AID that lead to partial or reversible degradation may be advantageous to avoid loss-of-function effects. In addition, it is important to verify that the degron tag does not alter the protein or process under study, similar to other protein tags. We did not observe changes to protein dynamics after degron tagging, with the expected exception of induced endocytosis and lysosomal degradation of a transmembrane protein. Ubiquitination of degron-tagged reporters will also create or disrupt protein-binding sites, similar to other protein modifications. Thus, degron tags are useful for probing the topology of transmembrane proteins, but not necessarily for studying their intracellular trafficking.

We used degron tags to label and track structures for which conventional reporters are insufficient. For example, EVs are typically detected by the tetraspanin proteins on their surface, but tetraspanin content is heterogeneous among EV subpopulations[46]. By degron-tagging membrane-associated or transmembrane proteins, we were able to specifically label EVs in vivo, which has proven to be a valuable tool to screen for new regulators of EV budding[2]. Although our PH::ZF1 reporter is likely to favour plasma membrane-derived EVs (microvesicles), it is possible to target endosome-derived EVs (exosomes) by degron-tagging proteins associated with the endosome surface. Alternatively, exosomes may also be labelled by degron-tagging transmembrane proteins, given that ubiquitination drives endocytosis of transmembrane proteins and relocalization to intraluminal vesicles (Supplementary Fig. 2i). Degron-tagging reporters found at both the plasma membrane and endosome surface, such as actin-binding domains or abundant proteins like GAPDH, should label both microvesicles and exosomes[47]. Thus, degron-tagged reporters are uniquely able to specifically label EVs, enabling in vivo tracking and functional studies.

Widefield light microscopy is normally limited to detecting structures that are >200 nm away from each other[48]. At this resolution, neighbouring membranes cannot be distinguished. In addition to specifically labelling EVs next to the plasma membrane, we showed that degron-tagged reporters could distinguish the cargo corpse membrane from the engulfing phagosome membrane. This enabled the visualization of corpse membrane dynamics during phagolysosomal clearance using widefield microscopy[20]. These reporters enable the precise staging of phagosomes for approaches such as correlative light and electron microscopy (CLEM)[49], which can be used to determine the ultrastructure of membrane breakdown during phagolysosomal clearance. Therefore, degron tagging is a useful tool to reveal novel insights into organelle dynamics in addition to the long-term tracking of specific cells in vivo.

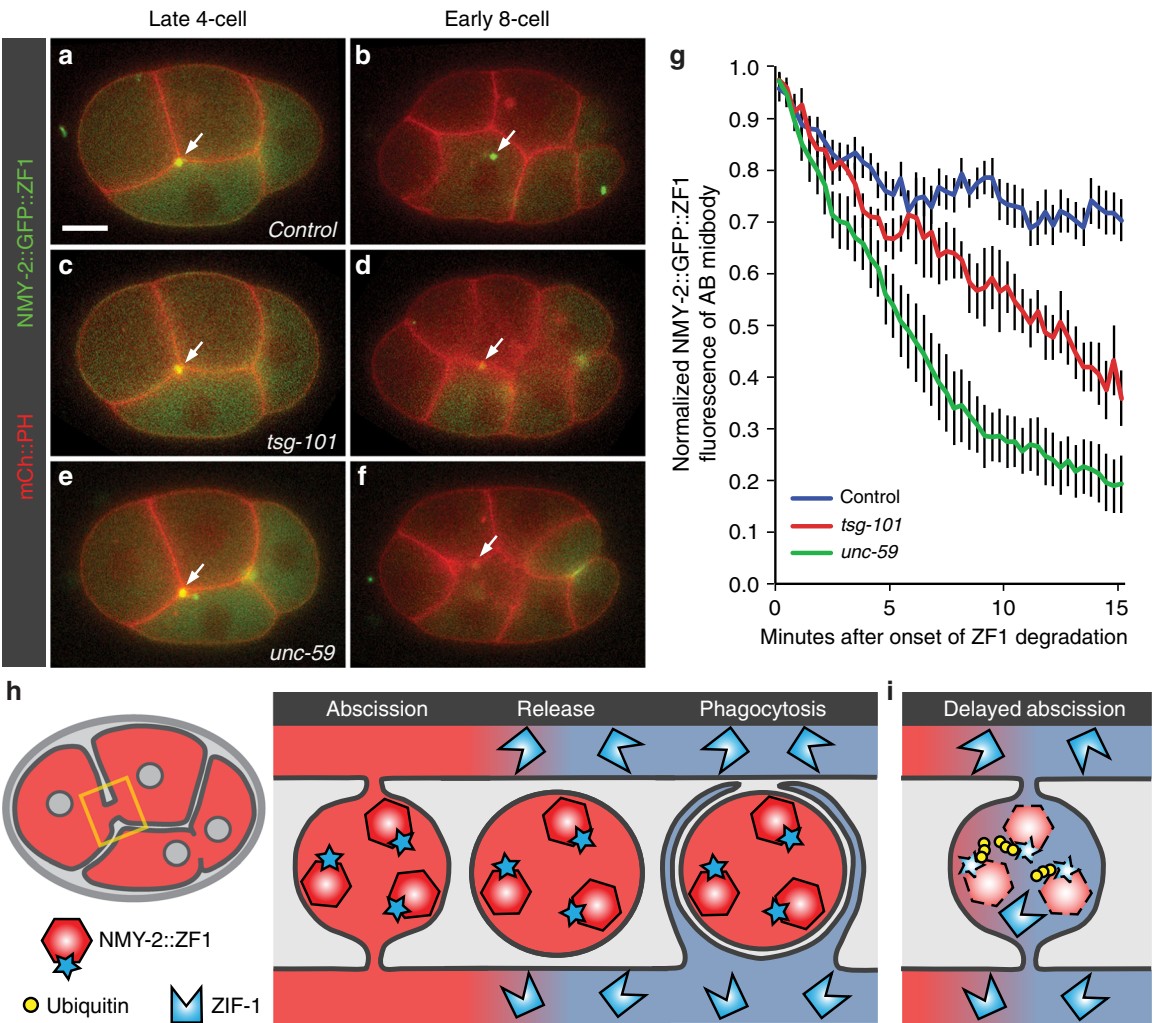

**Fig. 9** Degron reporters reveal membrane topology during abscission. **a**–**f** Time-lapse images of embryos expressing mCh::PH to label the plasma membrane and NMY-2::GFP::ZF1 to label non-muscle myosin in the cytokinetic ring. Scale bar: 10 μm. **a** In a control embryo, NMY-2::GFP::ZF1 between the anterior daughter cells (arrow) is protected from ZF1-mediated degradation due to its release outside cells in the midbody remnant after abscission. **b** NMY-2::GFP::ZF1 in the phagocytosed midbody remnant is protected from proteasomal degradation by engulfing membranes ($n = 19$). **c** In embryos depleted of the ESCRT-I subunit TSG-101, NMY-2::GFP::ZF1 localizes to the intercellular bridge normally. **d** NMY-2::GFP::ZF1 is degraded after *tsg-101* knockdown, indicating that abscission is incomplete and NMY-2::GFP::ZF1 is accessible to the degradation machinery. Engulfment of the AB midbody is also delayed, likely due to incomplete abscission ($n = 7$). **e, f** Embryos depleted of the septin UNC-59 show rapid degradation of NMY-2::GFP::ZF1 from the intercellular bridge and delayed engulfment of the midbody remnant due to defects in abscission ($n = 8$). **g** Fluorescence intensity of the NMY-2::GFP::ZF1 reporter on the intercellular bridge between anterior daughter cells (AB midbody) drops significantly for 2 min after the onset of ZF1 degradation in the cytoplasm of control embryos (blue, $n = 11$, $p < 0.05$ using Student's *t*-test with Bonferroni correction), showing that NMY-2 is able to diffuse out of the bridge. Fluorescence then persists, showing that formation of a diffusion barrier, symmetric abscission, and engulfment protect the degron-tagged reporter from degradation. In contrast, NMY-2::GFP::ZF1 fluorescence continues to drop in *tsg-101* RNAi-treated embryos (red, $n = 5$, $p < 0.05$ compared to control after 8 min) and *unc-59* mutants (green, $n = 6$, $p < 0.05$ compared to control after 3.5 min), showing that the ZF1 degron technique is sensitive to detecting small defects in abscission. Bars represent mean ± s.e.m. Source data are provided as a Source Data file. **h** In control embryos, degron reporters in the intercellular bridge are protected from proteasomal degradation due to release after abscission and encapsulation in a phagosome. **i** In abscission mutants, degron reporters are accessible to the cytosol, leading to their removal by proteasomal degradation

Using a degron-tagged reporter, we were able to measure early changes in the topology of the nuclear envelope during cell division as well as in the restricted space of the intercellular bridge, which was previously only possible using electron microscopy[36]. We were also able to detect defects in abscission after knocking down an ESCRT subunit[35], which were not visible using cytosolic diffusion assays[34]. Degron protection is therefore a sensitive tool to study the topology of membranes while they undergo fusion or fission.

Degron protection also revealed protein dynamics and topology. Knocking down septins or TSG-101 led to distinct rates of degron-mediated degradation of the myosin reporter, which could indicate that the intercellular bridge is open to differing degrees in these mutants, resulting in different rates of diffusion out of the bridge. We also found that a pool of the degron-tagged lamin reporter underwent degradation during interphase. This may indicate that an undetected level of ZIF-1 is in the nucleus, which ubiquitinates the ZF1 degron during interphase for degradation by the nuclear proteasome[50]. Alternatively, this may be due to undetected mobility of nuclear lamins into the cytosol during interphase. As cytosolic ligases are known to degrade proteins exported from the ER, while ER contents are protected

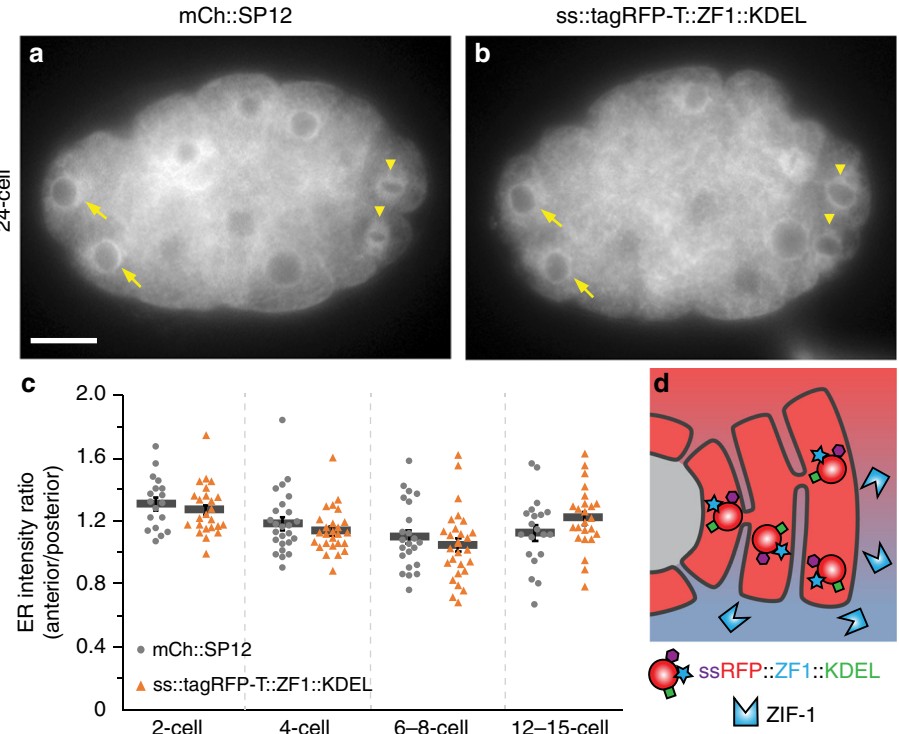

**Fig. 10** A ZF1 degron reporter in the ER is protected from ZIF-1/ECS-mediated degradation. **a** mCherry-tagged SP12 localizes to the endoplasmic reticulum (ER) in anterior (somatic, arrow) and posterior (germ line, arrowhead) cells ($n = 102$). Scale bar: 10 μm. **b** An RFP- and ZF1 degron-tagged KDEL reporter similarly localizes to the ER in both anterior (somatic, arrow) and posterior (germ line, arrowhead) cells, despite expression of the ZIF-1 ligase adaptor in anterior cells ($n = 88$). **c** There was no significant difference in the ratio between the fluorescence of anterior AB lineage cells and of posterior germ line or germ line sister cells at the indicated stages ($p > 0.05$, Student's $t$-test with Bonferroni correction, $n = 18$–25 mCh::SP12, $n = 24$–28 ss::tagRFP-T::ZF1:: KDEL). Bars represent mean ± s.e.m. Source data are provided as a Source Data file. **d** The ER membrane protects luminal proteins from ubiquitination by cytosolic ligase complexes

from ECS- or SCF-mediated degradation (Fig. 10)[51], degron-tagged reporters may also be used to reveal information on protein import/export across organelle membranes. As degradation of a reporter depends on its orientation as well as its location within the cell, degron protection assays can be applied to determining the topology of transmembrane proteins (type I/II) by testing whether they are accessible to degron-induced endocytosis and degradation, similar to our observations with SYX-4. Similarly, degron protection assays can distinguish cytosolic from luminal proteins. In summary, degron-tagged reporters improve the signal-to-noise ratio, reveal super-resolution insights on a standard microscope, and provide insights into localization and dynamics from the level of cells to proteins.

## Methods

**Worm strains and maintenance.** *Caenorhabditis elegans* strains were maintained according to standard protocol at room temperature or 25 °C[52]. For a list of strains used in this study, see Supplementary Table 1. *unc-59* loss-of-function mutant embryos were generated by feeding *unc-59* RNAi to the WEH132 strain bearing a hypomorphic *unc-59* mutation[35].

**Mammalian cell culture.** Cells were cultured according to standard mammalian tissue culture protocols including testing for mycoplasma. HeLa Kyoto cells (RRID: CVCL1922) and HeLa FRT/TO cells were a kind gift from Jonathon Pines (ICR, London, UK). HeLa cells were maintained in Dulbecco's modified Eagle's medium (DMEM) (Gibco) supplemented with 10% foetal bovine serum (FBS), 1% (v/v) penicillin–streptomycin, 1% (v/v) Glutamax and 0.5 μg/ml Amphotericin B. A Neon Transfection system (Thermo Fisher Scientific) was used to generate HeLa FRT/TO NLS-TIR1_P2A_NES-TIR1 cells[33] stably expressing Venus-mAID-LMNA. Stable single integrants were selected with 0.5 μg/ml puromycin (NLS-TIR1_P2A_NES-TIR1) and 200 μg/ml hygromycin (Venus-mAID-LMNA). For transient transfection with pCS2+_Flag-myc-NES-TIR1 and pIREShygro-Venus-mAID-LMNA, plasmids were co-electroporated into HeLa Kyoto cells according to

the manufacturer's instructions and seeded into 10 cm cell culture dishes. After 24–36 h, 2.5 mM thymidine (Sigma-Aldrich) was added to pre-synchronize cells at the border of G1 to S-phase. After 20 h of thymidine arrest, cells were washed, split into 96-well plastic bottom plates (μclear; Greiner Bio-One), and released into fresh media containing 332 nM nocodazole (Sigma-Aldrich) to ensure that cells remain in mitosis due to mitotic checkpoint activation during Venus-mAID-LMNA degradation. Stable HeLa FRT/TO NLS-TIR1_P2A_NES-TIR1/Venus-mAID-LMNA cells were pre-synchronized by the same protocol.

**Plasmid construction.** To generate pAZ132-coPH-oma-1 (219–378), pGF06 (ref. [2]) was amplified with primers oma-1 (378) Stop attL2 F and oma-1 (219) coPH R, while a C-terminal fragment of *oma-1* genomic DNA (aa 219–378) was amplified with primers coPH oma-1 (219) F and attL2 Stop oma-1 (378) R. Primer sequences are given in Supplementary Table 2. The PCR fragments were assembled using Gibson Assembly Mix (NEB) and pDonr221-coPH-oma-1(219–378) was then recombined into pAZ132-Gtwy (gift of Barth Grant) using Gateway cloning (Invitrogen).

To generate ss-pCFJ1954-KDEL, the signal sequence of *sel-1* was codon-optimized[53] and cloned into pCFJ1954 using around-the-world PCR with primers sel-1 ss flex F and sel-1 ss eft-3p R followed by a KLD reaction (NEB). A codon-optimized ZF1 domain from *pie-1* and the KDEL sequence were then cloned into ss-pCFJ1954 using around-the-world PCR with primers coZF1 KDEL Stop attB2 F and coZF1 flex R followed by a KLD reaction.

To generate pIREShygro-Venus-mAID-LMNA, the LMNA open reading frame was amplified from a plasmid encoding GFP-Lamin A[54] and cloned onto the C-terminus of 3xHA-Venus-mAID[33] within a pIRES-hygro3 backbone (Clontech). To localize TIR1 to the cytoplasm, pCS2+_Flag-myc-NES-TIR1 was used (Addgene plasmid #117717)[33]. To localize TIR1 to both the nucleus and cytoplasm, pCAGGs-NLS-TIR1_P2A_NES-TIR1 was used (Addgene plasmid #117699)[37].

**Worm transformation.** FT205, WEH251, WEH399, WEH434, and WEH447 were made by biolistic transformation using a Bio-Rad PDS-1000 according to the standard protocol[55]. The DP38 strain was bombarded with MP322 (gift of Michael Glotzer[21]) to generate FT205. WEH251 was generated by co-bombardment of pGF04 (ref. [20]) and pJN254 (gift of Jeremy Nance[56]) into the HT1593 strain. HT1593 was bombarded with pGF13 (ref. [20]) to generate WEH399, with pAZ132-

coPH-oma-1 (219–378) to generate WEH434, or with ss-pCFJ1954-KDEL to generate WEH447.

**RNAi experiments**. RNA interference (RNAi) was performed by feeding worms double-stranded RNA (dsRNA)-expressing bacteria from the L1 larval stage through adulthood at 25 °C (60–70 h) according to established protocols[57], except tat-5 RNAi was sometimes fed starting from the L3/L4 stage for 16–24 h. The following RNAi clones were used from available libraries (Source BioScience): tat-5 (JA:F36H2.1) and unc-59 (mv_W09C5.2), and previous studies: zif-1 (F2)[35]. The tsg-101 dsRNA was transcribed using T7 RNA Polymerase (ThermoFisher) from T7 PCRs of the tsg-101 RNAi plasmid(mv_C09G12.9)[35,58]. In all, 1 or 2 mg/ml tsg-101 dsRNA was injected into the gonad of young adult worms 20–26 h before analysis. Efficiency of tsg-101 RNAi was judged by a mild delay in internalization of the AB midbody remnant.

**Light microscopy**. Worm embryos were dissected from gravid adults and mounted in M9 buffer on an agarose pad on a glass slide. For imaging FT205 and FT368, Z-stacks were acquired on a Zeiss AxioImager, ×40 1.3 NA objective, an Axiocam MRM camera, and AxioVision software[21]. For N2, FT205, FT368, and XA3502, Z-stacks were acquired for GFP and DIC every minute at room temperature using a Leica DM5500 widefield fluorescence microscope with an HC PL APO ×40 1.3 NA oil objective lens supplemented with a Leica DFC365 FX CCD camera controlled by LAS AF software. For BV113, WEH02, WEH51, WEH132, WEH142, and WEH248, Z-stacks were acquired sequentially for GFP and mCherry every 20 s (refs. [20,35]). For WEH251, Z-stacks were acquired for mCherry every 20 s or every minute. For strains WEH260 and WEH296, Z-stacks were acquired for mCherry every minute[20]. For WEH399, Z-stacks were acquired for mKate2 and DIC every 30 s. For WEH434, Z-stacks were captured for mCherry every 40 s. Embryos that arrested during imaging were excluded from analysis. Time-lapse series were analysed using Imaris (Bitplane).

For mammalian cell imaging, a modified DMEM containing 10% (v/v) FBS, 1% (v/v) penicillin–streptomycin, 1% (v/v) Glutamax, and 0.5 µg/ml Amphotericin B without phenol red or riboflavin was used to reduce autofluorescence[59]. DNA was labelled by SiR-Hoechst (Spirochrome) at a final concentration of 50 nM 1–2 h prior to imaging. Cells were imaged for 6 h every 5 min by automated time-lapse microscopy on an ImageXpress Micro XLS widefield screening microscope (Molecular Devices) equipped with laser-based autofocus, ×10/0.5 numerical aperture (NA) and ×20/0.7 NA air objectives (Nikon), a Spectra X light engine (Lumencor), a sCMOS (Andor) camera (binning = 1 or 2), filters for YFP, Texas Red and Cy5, and a stage incubator at 37 °C in a humidified atmosphere of 5% $CO_2$. Auxin-dependent degradation was initiated at the beginning of imaging by adding 1-naphthaleneacetic acid sodium salt (NAA, ChemCruz) to a final concentration of 0.5 µM. Cells that died during imaging were excluded from analysis.

**Tracking**. H2B-labelled nuclei were tracked over time using the surface function of Imaris with thresholding to segment objects.

**Quantification of corpse membrane breakdown**. Corpse membrane topology was measured using a line scan across the second polar body phagosome in the WEH434 strain. A line with 3-pixel thickness was drawn through the middle of the phagosome using Fiji (NIH) and the mean profile intensity was measured.

**Fluorescence intensity measurements**. Mean fluorescence intensity of the mCh::PH::CTPD reporter (Fig. 3) was measured in a circle with an area of 0.5 µm² using ImageJ (NIH). Fluorescence intensity of the plasma membrane at the anterior side of the embryo near the first polar body was measured. Fluorescence intensity of the first polar body was measured as an internal control to correct for bleaching. Data are reported as the ratio of the fluorescence intensity of the plasma membrane to that of the polar body.

Mean fluorescence intensity of the ZF1::mCh::CHC-1 reporter (Fig. 4) was measured from images of four- and six-cell embryos. GFP::ZF1::PH fluorescence was used to trace a 5.48 µm long two-pixel width line along the plasma membrane using Fiji (NIH). Cell contacts between the anterior (ABa/ABp) or posterior (EMS/P2) cells were measured on the plasma membrane, in addition to the cytoplasm in both neighbouring cells. Data are reported as the mean intensity of the plasma membrane divided by the mean intensity of the cytoplasm.

Mean fluorescence intensity of the LMN-1 reporter (Fig. 7) was measured in a circle with an area of 0.8 µm² using ImageJ (NIH) in AB and later in ABp and its daughter nuclei as well as in P1 and later in EMS nuclei. Fluorescence intensity was measured until the LMN-1 marker in ABp daughter nuclei reappeared in zif-1 RNAi-treated embryos. A similar time range was used to measure embryos not treated with RNAi. Fluorescence intensity of the P1 and EMS nuclei was measured as an internal control. Data are reported as the ratio of the fluorescence intensity of the AB lineage nuclei to that of the P1 lineage nuclei. The control ratio was normalized to the zif-1 RNAi ratio to observe changes due to ZIF-1-mediated degradation. A line was fit to three parts of the curve in Excel to calculate the slopes of the sharp drops during mitosis (from −1 to 2.7 min after the start of cytokinesis

in the AB cell and from 11.3 to 13 min for ABx division) and the mild fluorescence loss during interphase (4 to 11 min).

For analysis of Venus-mAID-LMNA degradation (Fig. 8), all images were background-corrected with Fiji using a rolling ball radius of 50 and 100 pixels for images derived from ×10 and ×20 objectives, respectively. Subsequently, average Venus-mAID-LMNA fluorescence intensities of single cells were measured manually with Fiji, corrected for the remaining background by measuring image regions without cells and smoothed (four neighbours each size, fourth polynomal order) with Prism 6 (Graphpad). Smoothed intensities of Venus-mAID-LMNA were normalized to the beginning of the time-lapse ($t = 0$) to monitor degradation during interphase directly after the addition of NAA or normalized to NEBD to assess mitotic degradation. As transient transfections with NES-TIR1 and Venus-mAID-LMNA did not result in transfection of all cells with both constructs, only cells that expressed Venus-mAID-LMNA and showed degradation after NEBD were considered as double positive and analysed further (Fig. 8a, c, g, i). Average velocities of Venus-mAID-LMNA degradation were calculated during the first 45 min of the time lapse for interphase HeLa FRT/TO cells stably expressing NLS-TIR1_P2A_NES-TIR1 and Venus-mAID-LMNA or during 45 min before NEBD (interphase) and 45 min after NEBD (mitosis) for HeLa cells transiently expressing NES-TIR1 and Venus-mAID-LMNA.

Mean fluorescence intensity of the NMY-2::GFP::ZF1 reporter (Fig. 9) was measured in a circle with an area of 0.5 µm² using ImageJ (NIH)[35]. Midbody fluorescence was measured from contractile ring closure until the end of the time-lapse series or until the midbody was not distinguishable from the cytoplasm. Fluorescence intensity of the first polar body was measured as an internal control. An exponential decay curve was fit to the polar body data using OriginPro (OriginLab) and used to correct for fluorescence loss due to photobleaching. Embryos were excluded if the P0 and AB midbodies were too close to each other ($n = 4$) or if the polar body data did not fit an exponential decay function ($n = 1$). Embryos who arrested development during time-lapse imaging were excluded from measurements ($n = 3$). Embryos treated with tsg-101 RNAi were excluded if the AB midbody remnant internalized by the six-cell stage ($n = 6$), as the RNAi was judged ineffective. NMY-2 data are reported as the ratio of the fluorescence intensity of the midbody to the expected value of the polar body after cytoplasmic background subtraction. Timing of the onset of degradation of NMY-2::GFP::ZF1 in the cytoplasm was judged by eye by comparing the relative brightness of ABp and EMS using Imaris.

Mean fluorescence intensity of mCh::SP12 and ss::tagRFP-T::ZF1::KDEL (Fig. 10) were measured in a 5.4 µm² circle in 2- to 15-cell embryos using Fiji (NIH). Measurements were taken at the brightest regions near the nucleus in anterior AB lineage cells and posterior P lineage cells. Data are reported as a ratio of the mean fluorescence of the ER in the anterior AB lineage to the posterior P lineage. Images were excluded when the cell identity could not be determined ($n = 7$).

**Image processing**. For clarity, images were rotated, colorized, and the intensity was adjusted using Adobe Photoshop. All images show a single optical section (Z), except for Figs. 5, 7, 10 and Supplementary Figs. 2 and 4. In Fig. 5, six Zs with 1.2 µm steps were maximum projected using Leica LAS X software. In Fig. 7, three Zs with 1.2 µm steps were maximum projected using Leica LAS X. In Fig. 10, two Zs with 1.2 µm steps were maximum projected using Fiji (NIH). In Supplementary Fig. 2, images where maximum projected to span a region of 2.5 µm using Fiji (NIH). In Supplementary Fig. 4a, four Zs with 1.2 µm steps were maximum projected using Fiji (NIH). In Supplementary Fig. 4c, two Zs with 1.2 µm steps were maximum projected using Fiji (NIH). In Supplementary Fig. 4e, six Zs with 1.2 µm steps were maximum projected using Fiji (NIH). Figures 5c, f and Supplementary Movies were rotated, colorized, and the intensity was adjusted using Imaris. Several Zs were maximum projected in Imaris for Supplementary Movies. For the left movie in Supplementary Movie 4, the brightness was adjusted in each frame using Photoshop to compensate for photobleaching.

**Statistical evaluation**. Student's one-tailed t-test was used to test statistical significance with a Bonferroni correction to adjust for multiple comparisons when indicated. For Fig. 8, Prism 6.0 (Graphpad) was used for statistics and to create graphs. Venus-mAID-LMNA degradation curves represent single cell data of the indicated number of cells. Differences in average Venus-mAID-LMNA degradation velocities were analysed for significance using an unpaired multi-comparison Kruskal–Wallis test with Dunn's statistical hypothesis testing presenting multiplicity-adjusted p values. All experiments are representative of at least three independent repeats unless otherwise stated. No randomization or blinding was used in this study.

**Biological material availability**. All worm strains, cell lines, and plasmids used in this study are available from the authors or the indicated sources.

**Reporting Summary**. Further information on research design is available in the Nature Research Reporting Summary linked to this article.

## Data availability

The source data underlying Figs. 3c, 4e, 7k, 7l, 8g, 8h, 8i, 9g, 10c and Supplementary Figs. 1a–f, 3, 4b, 4d, 4f are provided as a Source Data file. Full data sets generated during and/or analysed during the current study are available from the corresponding author upon request.

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

## Acknowledgements

Michaela Geisenhof and Alida Melse provided technical assistance. The imaging facility of the Rudolf Virchow Center and the Light Microscopy Facility of the BIOTEC at TU Dresden provided support for imaging and data analysis. Strains and reagents were provided by Peter Askjaer, Zhirong Bao, Michael Glotzer, Barth Grant, Jeremy Nance, Karen Oegema, Jonathon Pines, Christian Pohl, and the *Caenorhabditis* Genetics Center (CGC), which is funded by NIH Office of Research Infrastructure Programs (P40 OD010440). This work was funded by Deutsche Forschungsgemeinschaft (DFG) grants FA1046/3-1 to G.F. and WE5719/2-1 to A.M.W., and European Research Council (ERC) Horizon 2020 Research and Innovation Program grant 680042 to J.M. We thank Avital Rodal, Cassie Blanchette, Katrin Heinze, Sonja Lorenz, and Anna Liess for critical reading of this manuscript.

## Author contributions

A.M.W. and J.M. conceived, designed, and supervised the study. K.B.B., G.F., K.J., L.I., J.C. and A.M.W. performed and interpreted experiments. K.B.B., G.F. and A.M.W. wrote the manuscript.

## Additional information

**Competing interests:** The authors declare no competing interests.

