## [Peer Review File · Nature Communications]

Reviewers' comments:

Reviewer #1 (Remarks to the Author):

The manuscript submitted by Beer et al. reports a "novel" approach to visualize certain specific membrane compartments within cells and organisms. Using the ZF1 degron in *C. elegans* embryos, the authors observed that when trying to degrade a membrane-localized PH-domain ZF1 reporter, two corpses within the *C. elegans* embryo remained labeled after ZF1-mediated degradation was initiated during the 4-cell stage when ZIF-1 started to be expressed. It turned out the two corpses were the two polar bodies, and in these bodies the PH-tagged ZF1 degron was protected from degradation because ZIF-1 is expressed in the zygote and cannot cross the membrane to degrade the target protein in the polar bodies. These results gave them the idea that membrane-protected compartments can be visualized with this novel "method". They then go on and show that microvesicles and extracellular vesicles can be clearly detected via the novel degron-based method, when using a Syntaxin-4 fused to ZF1. In addition, other cellular compartments such as phagocytosed cargo, taking the second polar body as an example (using H2B-ZF1 fusion constructs), can be investigated. Using yet other degron-tagged proteins (Lamin-1 or non-muscle myosin-2 fused to ZF1), they show that membrane topology can be studied during nuclear membrane breakdown and during abscission.

In the discussion, the authors propose several novel applications for the degron tagging they describe. These examples exemplify the potential use of the novel method as "an important addition to the cell biologist's toolbox".

The paper goes all the way from a possibly surprising observation to describing a novel approach to a number of biological questions, making the point that protein degradation methods can be used for much more than just for loss-of-function studies.

The paper is sort of complicated to read (since the method is so novel), but I think the work is well done and deserves publication in *Nat Comm*. The authors should be a bit more precise about the future potential applications to this method, and think about even further uses of the concept (express degrons and the corresponding proteasomal adaptor in tissue-specific manner, etc.). The approach as described in the paper relies a bit on serendipitous observations and making use of the later.

Reviewer #2 (Remarks to the Author):

Comments to the Authors (Required):

Review of Beer et al., "Degron tagging to label membrane-wrapped objects and probe membrane topology"

Background relevant to the work:

This work characterizes and exploits aspects of a protein degradation method developed by the Nance lab, which takes advantage of an endogenous protein regulation pathway in *C. elegans*. The ZIF-1 protein is expressed in early embryos, where it acts as a substrate recognition adapter for an ECS E3 ubiquitin ligase complex. Together with the ubiquitously expressed components ELC-1, CUL-2, and RBX-1, ZIF-1 targets proteins carrying a ZF1 "degron" (an 36-amino acid peptide derived from the PIE-1 protein) for degradation by the proteasome. Nance et al. "repurposed" this system by expressing ZIF-1 ectopically in other tissues, and showed that it could target ZF1 degron-tagged targets for degradation. A variety of other degron-mediated protein depletion systems have also been developed for use in *C. elegans*, mammalian cells, and other experimental models.

Summary of the work:

Here, the authors report that while ZIF-1-mediated degradation of cytoplasmic proteins is efficient (as previously reported), ZF1-tagged proteins within specific membrane-bound compartments, including polar body nuclei and extracellular and phagocytic vesicles, are resistant to degradation in the early embryo. They exploit this selectivity as a tool to deplete cytoplasmic proteins, enabling visualization of the pool of fluorescent proteins remaining within these compartments. Visualization

of fluorescent protein labeled subcellular structures has always been challenged by the high noise-signal ratio due to the appearance of such fluorescent proteins also in the cytosol, which increases the background fluorescence. They present evidence that this strategy improves imaging capacity for certain membrane wrapped structures where the E3 ligase and/or proteasome are not accessible. They claim that this strategy can be used to label/track certain cells, cell fragments, organelles and vesicles.

Overall assessment:

The observation that ZIF-1-mediated degradation in the early *C. elegans* embryo is inhibited in membrane-bound compartments will be of interest to researchers using this specific degradation system, particularly to those in the area of membrane trafficking. However, the generality of the conclusions is likely to be quite limited, and this is inadequately acknowledged in the manuscript. The authors present their results in a way that suggests that they apply to "degron tagging" of proteins in general, but their work specifically analyzes only ZIF-1-mediated degradation in *C. elegans*, and only in the early embryo. It is certainly possible that ZIF-1 or its E3 ligase cofactors cannot access specific membrane-bound compartments during early development, but there is ample evidence that other E3 ligases and the proteasome can regulate proteins throughout the cell, with the notable exception of the mitochondrial interior. Notably, ZIF-1 is not conserved outside of nematodes, and this method is thus unlikely to be used in other experimental systems. The work would also potentially be more useful if the compartmentalization had been analyzed in other *C. elegans* tissues, and I suggest that the authors may wish to examine this issue. Alternatively/additionally, it would be helpful to examine the fate of ZF1-tagged proteins within other membrane-bound compartments, such as the ER and Golgi. The work could potentially be extended by other authors if they can spatially confine other E3 ligases in space and/or time. Unfortunately, this was not emphasized in this study. But the authors may argue that is beyond the scope of work.

Minor concerns:

1. The title and abstract of this manuscript do not clearly convey the major findings of the work. The author may want to revise those parts to make it more comprehensible.
2. Fig.1 m, the architecture of the embryo looks disordered. Another cell membrane marker should be used to show whether degradation of PH-ZF1 causes any changes in membrane morphology and cell-cell attachment.
3. Long term developmental rate and locomotion activity should be assessed for all of the strains used in this study, including LMN-1 and SYX-4.
4. Fig. 4, a *mKate2::LMN-1* embryo should be included as control. It is not clear to which extent the endogenous degradation pathway contributed to the degradation of LMN-1 during ABx cell division.
5. Fig.4, robust ZIF-1-mediated degradation of functional proteins in cytosol and/or nucleus may potentially affect the timing of changes in membrane topology and dynamics. This should be discussed.
6. Scale bars should be included in all images.

Reviewer #3 (Remarks to the Author):

In this manuscript, Beer et al. report on the development of a novel degradation system that improves the dynamic visualization of proteins included in the lumen of membrane-bound organelles and vesicles and that can also be used to probe the topology of protein localization and the integrity of the nuclear envelope.

Altogether the experiments reported are well done even though one may like more quantitative data. My main concern is the limited applications I can see from this approach. Increased interest for the study would probably arise if the authors would really solve a key outstanding question that have resisted to other approaches, apply these methods to a large collection of reporters and/or if the authors would show, as suggested at the end of their abstract, that it may be efficient in other species than *C. elegans*. Last but not least, I am not convinced that such a tagging method is not strongly impacting the dynamics and behavior of the tagged proteins. This is particularly true for integral membrane proteins. Ubiquitination of these proteins may induce their intracellular transport toward endocytic compartments, may create novel protein complexes and perturb proper dynamics, etc.

For all these reasons, I do not think that this study would be fitting in a large audience journal like Nature Communications.

Some additional points the authors may like to take in consideration:

- 1- It would be interesting to stain ZIF-1 to confirm the inverted correlation between expression of the ZIF-1 and of the degron-tagged reporter.
- 2- Inactivating protein degradation (using a small molecule or RNAi for example) to confirm that the tagged reporter protein is then stabilized would be strengthening the study.
- 3- It would be important to test the effect of the tagged reporter on its endogenous counterpart. This is particularly important for proteins like NMY-2 which work as a dimer or multimer. The effect of NMY-2::GFP::ZF1 expression on the stability of endogenous NMY-2 would be important to quantify as it may impact the mechanism studied by the authors.

We thank the reviewers for their constructive ideas, which have helped us to substantially improve our manuscript. The revised manuscript includes 8 new figures, 2 revised figures, and 5 new videos. Below are detailed responses to the original critiques:

Reviewer #1:

The paper is sort of complicated to read (since the method is so novel), but I think the work is well done and deserves publication in Nat Comm.

We have added an additional model figure 1 to better explain the different degron methods, added additional models to figures 4, 5, 8, 9, and S3 to explain the different model systems, and edited the text to improve readability.

The authors should be a bit more precise about the future potential applications to this method, and think about even further uses of the concept (express degrons and the corresponding proteasomal adaptor in tissue-specific manner, etc.). The approach as described in the paper relies a bit on serendipitous observations and making use of the later.

We have added an additional figure 4 demonstrating the application of degron protection to determining protein topology, as well as adding a new figure 8 showing how spatial control of the ligase adaptor can determine which pool of the protein will be degraded (i.e. nuclear vs. cytosolic). We have also expanded on our discussion of spatial control in the intro and discussion.

Reviewer #2:

The observation that ZIF-1-mediated degradation in the early *C. elegans* embryo is inhibited in membrane-bound compartments will be of interest to researchers using this specific degradation system, particularly to those in the area of membrane trafficking. However, the generality of the conclusions is likely to be quite limited, and this is inadequately acknowledged in the manuscript. The authors present their results in a way that suggests that they apply to “degron tagging” of proteins in general, but their work specifically analyzes only ZIF-1-mediated degradation in *C. elegans*, and only in the early embryo. It is certainly possible that ZIF-1 or its E3 ligase cofactors cannot access specific membrane-bound compartments during early development, but there is ample evidence that other E3 ligases and the proteasome can regulate proteins throughout the cell, with the notable exception of the mitochondrial interior.

This is a key point that we inadequately addressed in the original manuscript. To demonstrate the broader applicability of degron protection assays, we have added new data using two additional degrons in combination with SCF ubiquitin ligases: the *oma-1* C-terminal phosphodegrons (Fig. 3, 6, & Video 2) and the auxin-inducible degron (Fig. 8, Videos 5-7). Both degrons gave us similar results to the ZF1 degron, strengthening our claim that the approach is generalizable to other degrons.

One key finding in our new data was that by localizing the Tir1 ligase adaptor to the cytosol in mammalian cells, we could cause cytosol-specific degradation of an AID-tagged reporter (Fig. 8A-C, Videos 5-6). Localizing it to both the nucleus and cytosol resulted in degradation in both compartments (Fig. 8D-E, Video 7). The ability to localize degradation using heterologous expression of ligase adaptor-degron pairs can be extrapolated to other ubiquitin ligases that are found in other organelles. Thus, using degrons is not limited to cytosolic degradation, as we originally proposed, but could be applied more broadly in the cell. In addition to the new figures, we have emphasized this important point in the discussion.

Notably, ZIF-1 is not conserved outside of nematodes, and this method is thus unlikely to be used in other experimental systems.

The ZIF-1-ZF1 system has newly been shown to degrade proteins in zebrafish (Yamaguchi et al., Elife 2019), demonstrating that it can interact with vertebrate ECS ubiquitin ligases. We have added this reference to the discussion to demonstrate that this method could be used in other experimental systems.

The work would also potentially be more useful if the compartmentalization had been analyzed in other *C. elegans* tissues, and I suggest that the authors may wish to examine this issue.

To demonstrate that degrons can probe compartmentalization beyond the *C. elegans* embryo, we show that degron protection assays work in mammalian cells in a new figure 8.

Alternatively/additionally, it would be helpful to examine the fate of ZF1-tagged proteins within other membrane-bound compartments, such as the ER and Golgi.

We tested whether a ZF1-tagged KDEL reporter could be degraded in the ER/Golgi and found that it was not (Fig. 10), consistent with protection by the intervening membrane.

The work could potentially be extended by other authors if they can spatially confine other E3 ligases in space and/or time. Unfortunately, this was not emphasized in this study. But the authors may argue that is beyond the scope of work.

As discussed above, we have added a new figure 8 showing the effects of spatially confining the ligase adaptor Tir1 with NES or NLS tags and emphasized this in the discussion.

Minor concerns:

1. The title and abstract of this manuscript do not clearly convey the major findings of the work. The author may want to revise those parts to make it more comprehensible.

We have revised the title and abstract to make the major findings clearer.

2. Fig. 1 m, the architecture of the embryo looks disordered. Another cell membrane marker should be used to show whether degradation of PH-ZF1 causes any changes in membrane morphology and cell-cell attachment.

Cell-cell attachment is normal in the PH-ZF1 strain, which can be seen in Fig. 2f, where the germ cell is being pulled to the ventral surface of the embryo by the gastrulating endodermal precursors. The changes in membrane morphology and cell-cell attachment in Fig. 2m are because this is a *tat-5* knockdown embryo. Their cells are rounded and weakly attached due to the abundant extracellular vesicle release, which results in gastrulation defects, as demonstrated by the germ cell remaining at the posterior in Fig. 2m (also in Wehman et al., Curr Biol 2011). Because the plasma membrane-associated PH-ZF1 is degraded, the only labeling that persists is in the extracellular vesicles pooling between the poorly adherent cells. To clarify this point, we have added a *tat-5* label to all panels in Fig. 2h-m.

3. Long term developmental rate and locomotion activity should be assessed for all of the strains used in this study, including LMN-1 and SYX-4.

We have added developmental rate data for LMN-1 and SYX-4 to Fig. S1B and long-term developmental rate for all strains to Fig. S1A. Only the published YFP-LMN-1 strain shows a significant delay in development (Fig. S1B) or uncoordinated worms (Fig. S1F), correlating with the observed changes to nuclear morphology after lamin overexpression (Fig. S1C-D). Notably, if we disrupt LMN-1-ZF1 degradation using *zif-1* RNAi, the nuclear morphology is also disrupted in this strain (Fig. S1C, new Video 4), suggesting that degron-mediated degradation can keep overexpression in check.

4. Fig. 4, a mKate2::LMN-1 embryo should be included as control. It is not clear to which extent the endogenous degradation pathway contributed to the degradation of LMN-1 during ABx cell division.

To determine which changes in mKate2::ZF1::LMN-1 were due to ZIF-1 mediated degradation, we have added measurements of *zif-1* RNAi-treated embryos to Fig. 7. Although the LMN-1 fluorescence is variable during the cell cycle, these embryos show no total change in fluorescence after ABx division (Fig. 7G, 7I, Video 4), confirming that there is no significant degradation. We also measured YFP::LMN-1 in the new Fig. S4B, which also does not show degradation after ABx division. We normalized the mKate2::ZF1::LMN-1 measurements to their *zif-1* RNAi-treated counterparts in Fig. 7H to only show the contribution of ZIF-1-mediated degradation

5. Fig.4, robust ZIF-1-mediated degradation of functional proteins in cytosol and/or nucleus may potentially affect the timing of changes in membrane topology and dynamics. This should be discussed.

We added a paragraph discussing the caveats of degron tags and pointing out that partial or reversible degradation may be advantageous to avoid loss-of-function effects when tagging functional proteins. We also added a sentence to clarify that all experiments were performed in the background of wild type untagged protein.

6. Scale bars should be included in all images.

Scale bars have been added to all figures.

Reviewer #3:

Altogether the experiments reported are well done even though one may like more quantitative data.

We have provided more quantitative data on degradation timing throughout the manuscript.

My main concern is the limited applications I can see from this approach.

Increased interest for the study would probably arise if the authors would really solve a key outstanding question that have resisted to other approaches, apply these methods to a large collection of reporters and/or if the authors would show, as suggested at the end of their abstract, that it may be efficient in other species than *C. elegans*.

As no other markers exist that are specific for extracellular vesicles, the gold standard for identifying them is electron tomography, a labor-intensive approach. Fluorescently labeling extracellular vesicles is a key outstanding problem for the extracellular vesicle field (see MISEV2018), which we have solved using degron-tagged reporters. We have added additional explanation of this to the text.

We have added a new figure 4 explaining the application of degron protection to determining protein topology. With these data, we answer a question that was not tenable with other approaches, namely whether clathrin was enriched at the plasma membrane in mutant worms or released in extracellular vesicles that are overproduced in these mutant worms. Answering this question would have been challenging with proteomics, based on the inaccessibility of the embryonic EVs, as well as with immunoelectron microscopy, because of the large size of antibodies (+/- 30 nm) used in comparison to the neighboring structures.

We have also added a new figure 8 demonstrating that the degron protection assay can be performed in mammalian cells using a degron-ligase adaptor system that is used in a variety of organisms.

Last but not least, I am not convinced that such a tagging method is not strongly impacting the dynamics and behavior of the tagged proteins. This is particularly true for integral membrane proteins. Ubiquitination of these proteins may induce their intracellular transport toward endocytic compartments, may create novel protein complexes and perturb proper dynamics, etc.

To test whether degron tagging was changing protein dynamics, we performed FRAP assays on the LMN-1::ZF1 reporter. We did not see increased recovery with the reporter (Fig. S3), suggesting that the protein was as immobile as a previously studied YFP-tagged LMN-1 (Galy et al., Curr Biol 2006).

The change to integral membrane protein trafficking after ubiquitination is a feature of our system. As we showed in Fig. S2, integral membrane proteins with cytosolic degrons are endocytosed and degraded. An integral membrane protein with an extracellular or luminal degron would be protected from ubiquitination, endocytosis, and degradation, allowing degrons to be used to determine protein topology. We have edited the text to more clearly state that ubiquitination of transmembrane proteins induces endocytosis and discuss the caveat that degron tags may not be useful for studying the intracellular trafficking of transmembrane proteins.

For all these reasons, I do not think that this study would be fitting in a large audience journal like Nature Communications.

We hope that the additional data demonstrating the wider applicability (two additional degrons and a mammalian model system) as well as the robustness of degron-tagging approaches addresses these concerns.

Some additional points the authors may like to take in consideration:

1- It would be interesting to stain ZIF-1 to confirm the inverted correlation between expression of the ZIF-1 and of the degron-tagged reporter.

We agree that this would have been an elegant way to demonstrate how the ZIF-1-ZF1 system works. We ordered the published ZIF-1 reporter strain to perform these experiments, but the transgene appears to have undergone epigenetic silencing, which we were unable to reverse. Since we added two additional degron systems to the paper, we added a new model figure 1 to clarify the published expression pattern of the ZIF-1 ligase adaptor as well as to explain how the other degrons work. We have also updated the text to clarify that it is primarily cytosolic.

2- Inactivating protein degradation (using a small molecule or RNAi for example) to confirm that the tagged reporter protein is then stabilized would be strengthening the study.

We performed *zif-1* RNAi to disrupt ZIF-1-mediated degradation of the LMN-1-ZF1 reporter and thereby demonstrate that the changes in reporter fluorescence are due to ZIF-1-mediated degradation. The reporter was stabilized after *zif-1* knockdown, and these data are added to Fig. 7G, 7I, and Video 4.

3- It would be important to test the effect of the tagged reporter on its endogenous counterpart. This is particularly important for proteins like NMY-2 which work as a dimer or multimer. The effect of NMY-2::GFP::ZF1 expression on the stability of endogenous NMY-2 would be important to quantify as it may impact the mechanism studied by the authors.

As antibodies to endogenous proteins would also recognize the degron reporters, we crossed the NMY-2-ZF1, H2B-ZF1, and LMN-1-ZF1 degron reporters to plain NMY-2, H2B, and LMN-1 reporter strains to demonstrate that untagged binding partners are not being degraded. We analyzed the fluorescence of the different markers in the presence and absence of the ZF1-degron reporters. In all three cases, we saw no effect on the plain reporters, which would suggest that the degron reporters do not generally impact endogenous proteins. These data are found in the new Fig. S4.

REVIEWERS' COMMENTS:

Reviewer #1 (Remarks to the Author):

I feel that the authors have responded to my remarks and points raised.

Reviewer #2 (Remarks to the Author):

Review of Beer et al., "Degron tagging to label membrane-wrapped objects and probe membrane topology"

The authors have now added several experiments to address my comments, which have expanded the scope of the manuscript. The revisions have addressed all of my major concerns that directly relate to the scientific content of this manuscript. While the generality of the observations is still quite limited, I believed that the strategy described in this work will benefit future studies involving labeling of membrane-wrapped compartments and/or probing of membrane topology in certain contexts. In my view, it is appropriate for publication in Nature Communications.

Reviewer #3 (Remarks to the Author):

This is a revision from a manuscript from Beer et al. that reports on the development of a novel degradation system that improves the dynamic visualization of proteins included in the lumen of membrane-bound organelles and vesicles and that can also be used to probe the topology of protein localization and the integrity of the nuclear envelope.

I find that the authors did a very good job answering my comments. In particular, it is now shown that it may be applied to various degradation system, including a regulated one and that it would work in various species. This opens up the system and may interest a larger audience. For these reasons, I think that the study can be accepted for publication in Nature Communications.

Specific comments:

- The authors tried to answer to my comments questioning the "gain of function effect" of their tagging system on the normal dynamics of proteins, in particular on endocytosis. They carry out FRAP experiment to show that the dynamics was not affected but is was not really my point My question was more to test whether tagging and ubiquitination would not change the behavior of the protein. For example, would a protein not naturally found internal vesicle be found there because of induced ubiquitination ? If the authors cannot carry out this experiment that should discuss this point to clarify the limit of their approach and the points to survey.
- The novel figure 8 is important but should be improved. The cells shown in 8d seem to be in a very bad shape, possibly dying (in particular the cell on the top). The authors should show an additional field unless the system is toxic and the authors should then state this and comment on it.
- Still about the novel I would select less frames in Fig8 a and b. They will appear very same in the final publication and such a large number of frames is not needed because quantification curves are provided (8f and g). I actually do not really understand why 8a and 8b are name as such since this in the same cell. I think it should be only 8a covering interphase and division with a few chosen frames.
- The authors show in figure 7G and 7I that RNAi of ZIF1 inhibits the degradation and hence that ZIF1 is specific for the degradation of ZF1-LMN1. However, the figure is a bit confusing. Instead of only showing one image (i+i') of RNAi, create a panel (like c-f +c'-f') for RNAi ZIF1 as to show the blockage of ZF1::LMN-1 degradation. This will also be more in line with data presented in 7G and 7H.

Response to REVIEWERS' COMMENTS:

Reviewer #1 (Remarks to the Author):

I feel that the authors have responded to my remarks and points raised.

Thank you for your constructive comments.

Reviewer #2 (Remarks to the Author):

Review of Beer et al., “Degron tagging to label membrane-wrapped objects and probe membrane topology”

The authors have now added several experiments to address my comments, which have expanded the scope of the manuscript. The revisions have addressed all of my major concerns that directly relate to the scientific content of this manuscript. While the generality of the observations is still quite limited, I believed that the strategy described in this work will benefit future studies involving labeling of membrane-wrapped compartments and/or probing of membrane topology in certain contexts. In my view, it is appropriate for publication in Nature Communications.

Thank you for your suggestions, which considerably improved this study. We hope this strategy will be of use for other groups.

Reviewer #3 (Remarks to the Author):

This is a revision from a manuscript from Beer et al. that reports on the development of a novel degradation system that improves the dynamic visualization of proteins included in the lumen of membrane-bound organelles and vesicles and that can also be used to probe the topology of protein localization and the integrity of the nuclear envelope.

I find that the authors did a very good job answering my comments. In particular, it is now shown that it may be applied to various degradation system, including a regulated one and that it would work in various species. This opens up the system and may interest a larger audience. For these reasons, I think that the study can be accepted for publication in Nature Communications.

We are glad that the additional experiments provided compelling evidence to support publication of this method.

Specific comments:

- The authors tried to answer to my comments questioning the “gain of function effect” of their tagging system on the normal dynamics of proteins, in particular on endocytosis. They carry out FRAP experiment to show that the dynamics was not affected but it was not really my point. My question was more to test whether tagging and ubiquitination would not change the behavior of the protein. For example, would a protein not naturally found internal vesicle be found there because of induced ubiquitination? If the authors cannot carry out this experiment that should discuss this point to clarify the limit of their approach and the points to survey.

We understood that you were asking two different questions and tried to address one experimentally and the second one with text changes. Ubiquitination, like any protein modification, certainly changes the behavior of the protein, in this case binding partners, half-life, and localization of the protein (Trafficking to proteasome or lysosome). Degrading reporter proteins from certain locations will transiently alter the behavior of reporter proteins in these locations. We have added a more direct reference to this in the discussion where we discussed potential caveats and suggestions for avoiding loss-of-function effects.

Syntaxins are normally targeted by ubiquitin ligases for endocytosis and lysosomal degradation (LS Chin et al., JBC 2002), which is considered the classical method for the turnover of post-Golgi transmembrane proteins (Piper & Luzio, Curr Opin Cell Biol 2007). Thus, we predict we are speeding up the normal degradatory pathway or decreasing the half-life of the protein. To clarify this, we added references to post-Golgi and intraluminal vesicles in the main text. The previous description and model figure in supplemental figure 2 could be easily overlooked by readers.

It would be interesting to examine what would happen to a degron-tagged ER transmembrane protein that is normally degraded by proteolysis in the ER membrane, but that is beyond the scope of this study.

- The novel figure 8 is important but should be improved. The cells shown in 8d seem to be in a very bad shape, possibly dying (in particular the cell on the top). The authors should show an additional field unless the system is toxic and the authors should then state this and comment on it.

HeLa FRT cell lines take longer to spread than HeLa Kyoto cells. We have replaced the cells in Figure 8e (formerly 8d) to more clearly show that this stable transgenic cell line is healthy.

- Still about the novel I would select less frames in Fig8 a and b. They will appear very same in the final publication and such a large number of frames is not needed because quantification curves are provided (8f and g). I actually do not really understand why 8a and 8b are name as such since this in the same cell. I think it should be only 8a covering interphase and division with a few chosen frames.

We have decreased the number of frames shown in the time lapse series in Fig. 8. We have labeled the different meanings of time 0 to more clearly demonstrate the different labeling between the interphase and mitotic cells, as well as reorganizing the figure and expanding one of the models.

- The authors show in figure 7G and 7I that RNAi of ZIF1 inhibits the degradation and hence that ZIF1 is specific for the degradation of ZF1-LMN1. However, the figure is a bit confusing. Instead of only showing one image (i+i') of RNAi, create a panel (like c-f +c'-f') for RNAi ZIF1 as to show the blockage of ZF1::LMN-1 degradation. This will also be more in line with data presented in 7G and 7H.

We have added additional images from the *zif-1* time-lapse to display it consistently with control embryos to Fig. 7g-j, as well as adding new model figures to Fig. 7m-n.